# Towards decoding individual words from non-invasive brain recordings

Stéphane d'Ascoli [1] ✉, Corentin Bel[2,3], Jérémy Rapin[1], Hubert Banville[1], Yohann Benchetrit[1], Christophe Pallier [2] & Jean-Rémi King[1,3] ✉

While deep learning has enabled the decoding of language from intracranial brain recordings, achieving this with non-invasive recordings remains an open challenge. We introduce a deep learning pipeline to decode individual words from electro- (EEG) and magneto-encephalography (MEG) signals. We evaluate our approach on seven public datasets and two datasets which we collect ourselves, amounting to a total of 723 participants reading or listening to five million words in three languages. Our model outperforms existing methods consistently across participants, devices, languages, and tasks, and can decode words absent from the training set. Our analyses highlight the importance of the recording device and experimental protocol: MEG and reading are easier to decode than EEG and listening, and decoding performance consistently increases with the amount of data used for training and for averaging during testing. Overall, our findings delineate the path and remaining challenges towards building non-invasive brain decoders for natural language.

In less than five years, artificial intelligence (AI) has been redefining the frontiers of brain-computer-interfaces (BCIs). Several groups have now demonstrated that intracranial implants in the motor cortex could be used to efficiently decode words from brain signals. For example, several machine learning algorithms can now learn to recognize patterns of neural activity associated with the intention to write or pronounce letters or syllables [1–8]. Such neuroprostheses could thus provide a voice to individuals who, after a brain lesion or a neurological disorder, have lost the ability to speak or communicate.

However, such a BCI requires an intracranial device and thus neurosurgery. In addition, cortical implants can be difficult to maintain beyond several months[9]. Non-invasive BCIs thus remain an important objective to assist or diagnose brain-lesioned patients[10–13].

Several non-invasive BCIs based on electro- or magneto-encephalography (M/EEG) exist but typically require users to perform tiring tasks over extended time periods, such as sustained visual attention or motor imagery[14]. For example, the P3-Speller[15] is a popular protocol where participants can spell individual letters by paying attention to flickering stimuli on a computer screen. However, these approaches are too slow and too demanding to effectively scale to the constraints of natural language.

This paradigm may be shifting, however. Over the past two years, two groups[16,17] independently proposed a similar solution to directly decode sentences from non-invasive brain recordings of participants listening to natural language, by learning to align brain activations to those of AI language models. However, each of these two studies faced distinct limitations.

Tang et al. [16] relied on functional Magnetic Resonance Imaging (fMRI) to guide a language model during its text generation process. Although fMRI provides a good spatial resolution of the brain, its low temporal resolution stands as a barrier: each brain image is affected by all the words occurring in a multi-second time window, preventing the precise decoding of individual words.

In contrast,[17] used EEG and MEG – fast neuroimaging devices, which have sufficient temporal resolution to access word-level. However, the authors used a *speech retrieval* strategy, where the decoder is trained to identify the most likely brain recording segment corresponding to a given speech segment. While this method achieves relatively good decoding performance (70% top-10 accuracy), it requires access to the ground truth speech segments at test time, which limits its practical relevance. Additionally, it is unclear whether the decoder relies on the perceptual

[1]Meta AI, Paris, France. [2]CNRS, INSERM, CEA, Neurospin center, Gif-sur-Yvette, France. [3]Laboratoire des Systèmes Perceptifs, École Normale Supérieure, Paris, France. ✉e-mail: sdascoli@meta.com; jeanremi@meta.com

characteristics of the speech segment or the semantic features of the underlying words.

Finally, the scalability and robustness of language decoding remain unknown: most studies focus on a unique device, a single task, and a few participants. In sum, decoding, at scale, individual words from non-invasive approaches remains an open challenge.

To tackle this issue, we propose a new model trained to decode individual words from EEG and MEG recordings (see Fig. 1). We validate our method with an unprecedentedly large dataset, encompassing 723 participants, either recorded with EEG or MEG while listening or reading sentences in their native language (see details in Table 1). Our pipeline achieves state-of-the-art decoding performance, with up to 37% top-10 accuracy with a retrieval set of 250 words (see some decoded sentences in Table 2).

## Results

### Decoding performance across models and protocols

**Linear models.** We first aim to verify that word embeddings can be decoded from M/EEG signals with a linear model[18–21]. For this, we train a ridge regression model to predict the word embeddings from a unique time sample of the M/EEG recording at a time $\tau$ relative to the corresponding word onsets, and vary $\tau$ between $-0.5$ s and 2.5 s. We evaluate the decoding performance by computing the Pearson correlation between each dimension of true and the predicted embeddings on the test set, then averaging across all dimensions. The resulting scores tend to peak within the first 500 ms, although with varying performances across datasets (Fig. 2A).

While ubiquitous in neuroscience, linear decoders may not be optimally designed to leverage the complex neural signals embedded in noisy M/EEG recordings. Indeed, a retrieval metric, based on the word embeddings linear models predict, leads to statistically significant ($p < 0.005$) but extremely low top-10 accuracies: e.g., around

6% for the best datasets, where the chance level is 4% (Fig. 2B). To address this issue, we now turn to the deep learning architectures.

**Deep learning models.** We compare three deep learning pipelines, each trained with the same contrastive objective: EEGNet[22],[17]'s "Brain-Module", and our pipeline. All three models significantly outperform linear decoders (Fig. 2B) ($p < 0.005$, paired Wilcoxon test).[17]'s Brain-Module, which is equipped with a subject layer, and thus designed to learn from different participants, significantly outperforms EEGNet ($p < 0.005$), with a twofold increase in accuracy on average over datasets. Finally, adding the transformer to operate at the sentence level yields another 50% performance boost on average over this BrainModule. This shows that our approach leads to a major improvement in comparison to existing M/EEG models.

The decoding performance of our pipeline is well above chance for each of the 773 subjects included in our study (Fig. 2C). Interestingly, the variability across subjects tends to be lower in the listening datasets than in the reading datasets: for example, for the LittlePrince datasets, accuracy varies between 26 and 33% in the listening condition, and between 19 and 38% in the reading condition.

Overall, these results highlight that our approach can reliably decode words from brain activity across a variety of participants, recording devices, tasks and languages.

**Impact of the experimental protocol.** How do the various aspects of the experimental protocol impact decoding performance? Mann-Whitney tests across participants show that our decoding pipeline performs better when subjects read rather than listen to sentences. Indeed, the pairwise comparison of results for the Schoffelen and LittlePrince datasets, where the reading and listening stimuli were the same, yields $p < 10^{-16}$ (Fig. 2E). Several reasons may explain this phenomenon. First, low-level visual features like word length are more

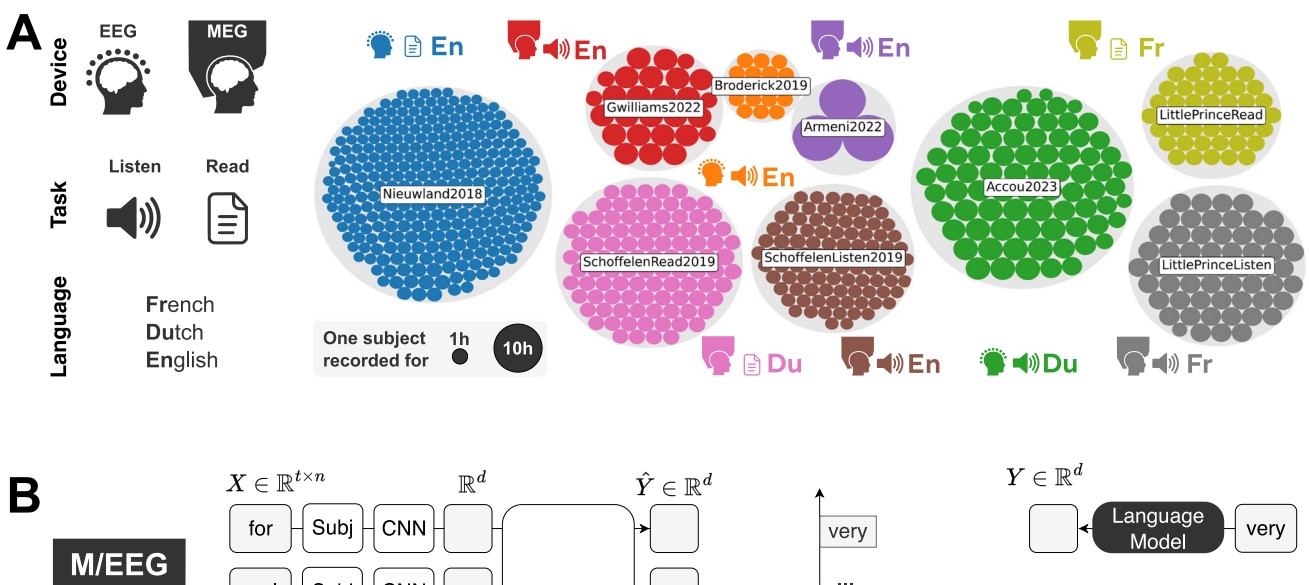

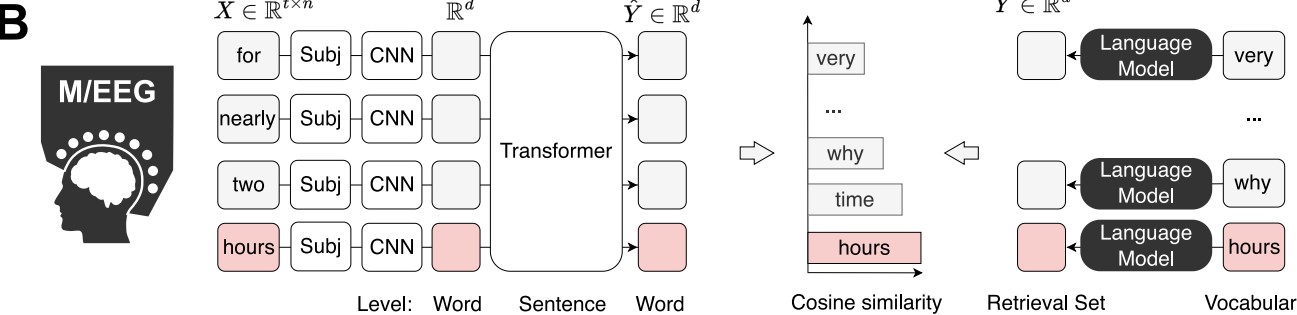

**Fig. 1 | Approach. A** Each colored disk represents 1 subject (size represents recording time). Our datasets encompasses both public and original M/EEG data of participants reading or listening to Dutch, English or French sentences (Table 1). **B** Our deep learning pipeline consists in training, with contrastive learning, an architecture that decodes the semantic representations of words from brain activity, as identified by a pretrained multilingual language model.

## Table 1 | Summary of the datasets considered

| Dataset | Subjects | Time (h) | Words (k) | Unique words (k) | Unique sentences (k) | Language | Task | Narratives | Device | Sensors |
|---|---|---|---|---|---|---|---|---|---|---|
| 28 | 295 | 171 | 706 | 0.9 | 0.3 | English | Read | False | EEG | 63 |
| 69 | 19 | 20 | 214 | 1.6 | 0.8 | English | Listen | True | EEG | 128 |
| 27 | 80 | 150 | 1255 | 13.5 | 15.2 | Dutch | Listen | True | EEG | 64 |
| 72 | 27 | 57 | 419 | 2.1 | 0.7 | English | Listen | True | MEG | 208 |
| 71 | 96 | 81 | 264 | 1.8 | 0.7 | Dutch | Listen | False | MEG | 273 |
| 71 | 99 | 106 | 271 | 1.8 | 0.8 | Dutch | Read | False | MEG | 273 |
| LittlePrinceListen | 58 | 94 | 874 | 2.4 | 1.5 | French | Listen | True | MEG | 306 |
| LittlePrinceRead | 46 | 59 | 623 | 2.6 | 1.3 | French | Read | True | MEG | 306 |
| 26 | 3 | 34 | 247 | 7.3 | 5.0 | English | Listen | True | MEG | 298 |
| Total | 723 | 772 | 4877 | | | | | | | |

## Table 2 | Examples of decoded sentences

| True words | Decoded words |
|---|---|
| Yes, it would be as well | Yes, it may be the guess |
| Sherlock Holmes sat up with a whistle | Sherlock Holmes stretched up on the wall |
| Here we are said Holmes cheerily as we filed into the room | Here her clock said Holmes rattling up the corner into the door |
| He included us all in a sweeping bow and stalked out of the room | She surveyed were sweeping in a hotel basket and limped out of the bridge |
| He pushed past the servant and rushed into the drawing room followed by the king and myself | He walked open the stair and brushed into the wooden stair pulling through the bridge of door |
| I am ashamed of you Holmes said, Lestrade with dignity after a few minutes' silence | If am sign of mr office said remarked said ejected after a few marked resource |

We report sentences from [26]'s dataset, with the original stimulus on the left, and the decoded sentence on the right.

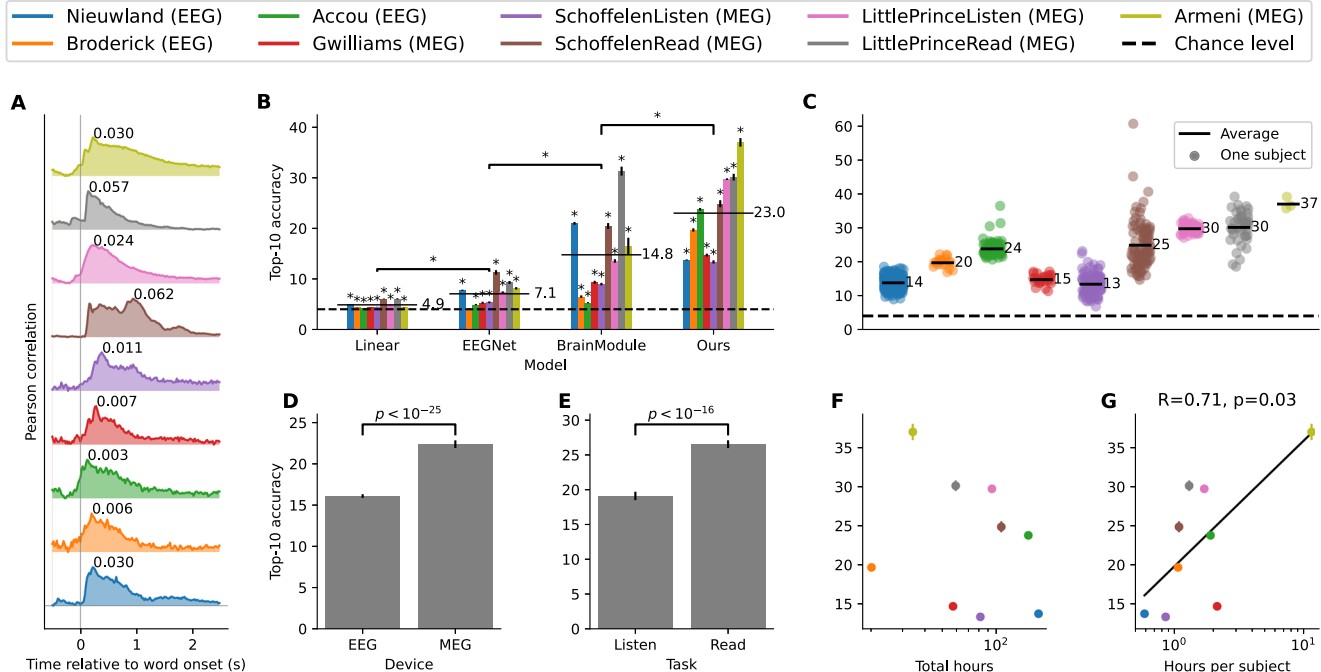

**Fig. 2 | Decoding performance across model architectures and datasets. A** A linear ridge regression is trained to predict word embeddings from a single slice of M/EEG data, and at each time sample relative to word onset. Predictive accuracy is evaluated on the test set as the Pearson correlation averaged over the embedding dimensions. We fit a different model for each subject, and report the average over all subjects. Each curve is normalized to its peak value, which is explicitly indicated above. **B** We compare the accuracy for classic decoding models for each dataset (colors). Horizontal black lines denote the average across datasets for a given model. Stars highlight above chance decoding across participants ($p < 0.005$, one-sided $t$-test). **C** Accuracy of our model for each subject of each dataset, with the average over subjects denoted as horizontal lines. **D** Accuracy averaged by recording device: MEG recordings are significantly better decoded than EEG recordings ($p < 0.005$, Mann-Whitney U-test). **E** Accuracy averaged by task. We focus on MEG datasets that had the same sentences in the reading and listening conditions: reading tasks are significantly better decoded than listening tasks ($p < 0.005$, Mann-Whitney U-test). **F** Accuracy compared to the total recording duration of each dataset. **G** Accuracy compared to the average recording duration per subject. The log-linear fit yields $p < 0.05$ (one-sided $t$-test), and the Spearman correlation and $p$-value are reported above the figure. In panels (**B–G**), the error bars represent the SEM across subjects (the number of subjects for each dataset is reported in Table 1).

readily represented and thus help decoding, as will be shown in what follows. Second, unlike the spoken words, which are not clearly segmented into low-level segments, the RSVP protocol may help isolate each word and thus improve our word-level decoding.

Comparing the recording devices, we observe that decoding performances are higher with MEG than EEG ($p < 10^{-25}$, Fig. 2D). This is unsurprising, given that MEG typically boasts higher signal-to-noise ratios. However, since we do not have access to datasets with identical stimuli for both recording devices, this comparison is less robust than the comparison of perception modalities and warrants further exploration.

Finally, while decoding accuracy does not consistently vary with the datasets sizes,(Fig. 2F), we observe a weak trend with the log volume of data *per subject* (Fig. 2G, $p < 0.05$). This suggests that with a fixed recording budget, one should favor *deep* datasets (few participants over many sessions) than *broad* datasets (many participants over few sessions). This result is consistent with other decoding studies based on fMRI[23,24], and suggests that current experimental paradigms, typically based on the repetition of stimuli across participants, may limit the diversity of the training data and could hinder the decoding scalability across broad datasets. Finally, it is important to note that collecting a lot of data from a few individuals may limit other goals, such as the possibility to train a model that could decode language from the brain activity of unseen participants.

Together, these results highlight the impact of experimental designs such as recording device, tasks and time allocation per participant.

### Scaling laws of word decoding

How does decoding performance scale with M/EEG data? To address this issue, we now focus on our decoding pipeline and analyse how its decoding performance varies with (1) the amount of training data, (2) the amount of test words used to average decoding predictions, and (3) the type of averaging used to improve decoding predictions.

**Scaling training data.** We retrain a series of decoding pipelines on different subsets of our dataset, obtained by gradually increasing the number of subjects, starting with a single subject. The results show that decoding performance increases with the amount of training data (Fig. 3A), following a roughly log-linear trend. This result, which mirrors the findings of[17], demonstrates the effectiveness of the subject layers: they enable our model to generalize across participants, at the cost of preventing zero-shot generalization to new participants (for

which the subject layers need to be fine-tuned). Although the trends vary across datasets, we do not observe clear signs of diminishing returns, hinting at the scalability of this decoding technique with the amount of experimental data.

**Scaling test data averaging.** All of the decoding metrics reported so far correspond to *single-trial* performances, such that they can be compared across datasets and inform real-time applications. However, studies often report decoding performances obtained from the *average* of multiple identical trials (e.g.[25]). While such averaging cannot be directly translated to real-time conditions, it may clarify whether decoding failures are solely caused by the low signal-to-noise ratio of M/EEG (in machine learning terms, a high variance) or also reflect an imperfect learning of the mapping (a high bias).

In our setup, decoding performance steadily increases with the number of decoding predictions used for averaging, following a very clear log-linear trend. Most datasets show a two-fold improvement with such a technique (Fig. 3B). Remarkably, for the dataset of[26], the top-10 accuracy reaches close to 80% after averaging only 8 predictions in response to the same word. These results suggest that decoding performance is strongly constrained by the low signal-to-noise ratio: reducing the latter via averaging improves performance drastically.

**Averaging across subjects or contexts.** In the previous analysis, we average the occurrences of words both over (i) repetitions for a given subject in response to different contexts and (ii) repetitions for several subjects in response to the same context (see Fig. 3C for an illustration). How do these two averaging methods compare with each other?

To answer this question, we focus on the[27] and LittlePrince datasets, which contain both a large number of subjects and a large amount of data per subject, and on the 50 most frequent words, which are repeated at least four times each for each subject. Our results show that decoding performance increases substantially more rapidly when averaging over contexts than over subjects (Fig. 3C). This explains why datasets which feature a large amount of sentences but few participants, such as that of[26] benefit more from averaging than those which feature a large amount of participants but few sentences such as that of[28] (Fig. 3B).

### Interpreting decoding performances

Our transformer model can decode individual words from brain responses to natural language across a variety of experimental conditions. Do the decoders rely on word semantics?

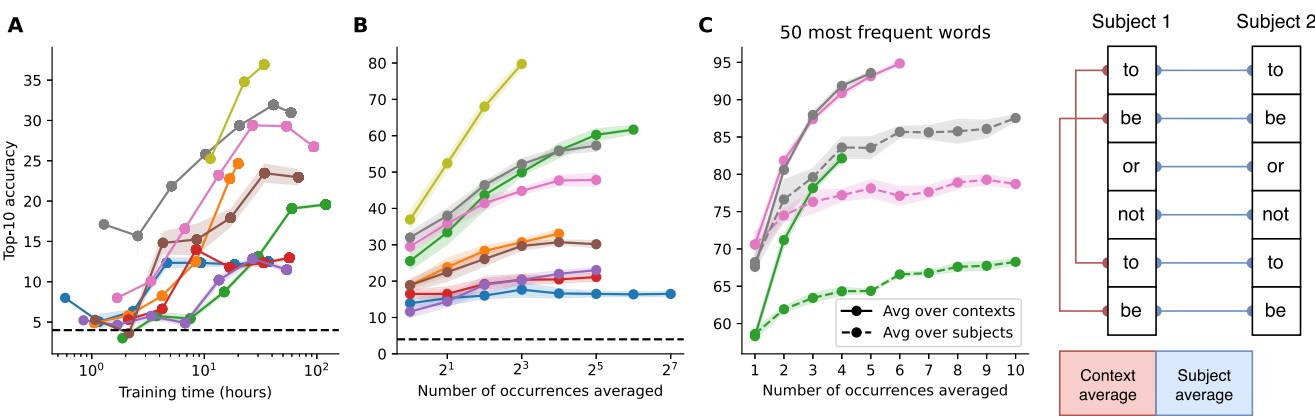

**Fig. 3 | Scaling laws for decoding performance. A** Balanced top-10 accuracy as a function of the number of subjects used for training. Shaded regions indicate SEM across the subjects. **B** Balanced top-10 accuracy as a function of the number of occurrences of each word averaged before scoring. Shaded regions indicate the standard deviation over the sampling of the occurrences. **C** Comparison of averages within a given context and across *N* subjects *versus* averages within a given

subject and across *N* contexts for that subject, as illustrated by the sketch on the right. As this necessitates both the presence of many repetitions across subjects and contexts, we focus on the Accou and LittlePrince datasets, which match these constraints, and consider the 50 most frequent words. Shaded regions indicate the standard deviation over the sampling of the occurrences.

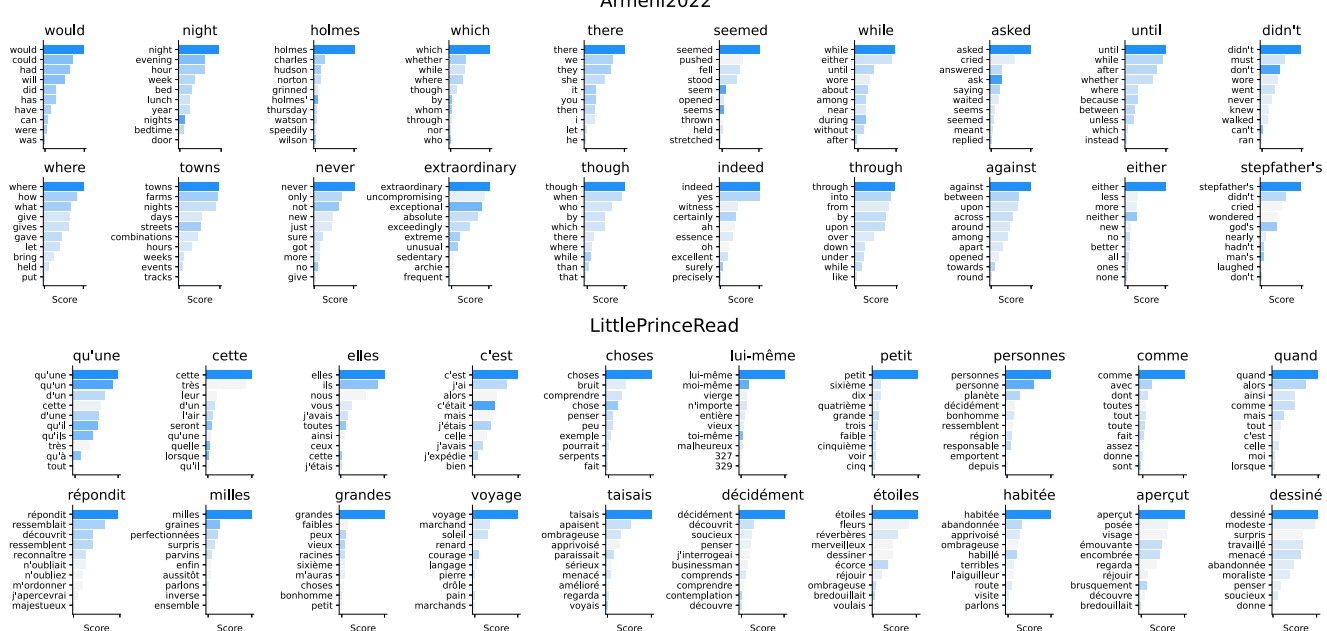

**Fig. 4 | Examples of top-10 predictions for two MEG datasets.** The y-axis indicates the 10 most likely words given MEG activity. Horizontal bars represent the words $Y_j$ with the highest cosine similarity to the decoding prediction $\hat{Y}_i$: $\mathbb{E}\left[\measuredangle(\hat{Y}_i, Y_j)\right]$, where $\mathbb{E}$ denotes an average over all the recordings in the test set corresponding to word $i$. The colorscale indicates the cosine similarity to the true word, i.e., $\measuredangle(Y_i, Y_j)$, with semantic similarity increasing from light to dark blue. Overall, many decoding predictions seem semantically similar to the words actually presented to the subjects. More examples are displayed in ?

**Analysis of decoding predictions.** To address this question, we first inspect the predictions of the decoder for individual words (Fig. 4, more examples can be found in Supplementary Fig. 6). For simplicity, we focus this qualitative analysis on[26] and LittlePrinceRead.

For both datasets, many predictions appear to often fall close to the meaning of the word actually presented to the participants. For example, for the word 'night', the top five predictions are 'night', 'evening', 'hour', 'week' and 'bed'. Given that word embeddings are known to capture semantic features[29], this result is coherent with the learning objective of our model.

For LittlePrinceRead, the predicted words appear to also capture visual features. For example, long words like 'décidément' tend to be decoded as long but semantically-unrelated words. Similarly, for words containing hyphens, apostrophes and accents, the top predictions often contain these special characters.

**Analysis of mistakes.** To quantitatively assess the relationships shared between the true words and their decoded predictions, we compare the incorrect top-1 predictions to their related true words (Fig. 5). We analyze four properties: whether the true and predicted words share the same first letter, the same last letter, whether they have the same number of letters, and finally, whether they share the same part-of-speech (*i.e.* whether the words are both nouns, verbs, determinants, etc.).

For almost all datasets, incorrect predictions share the same parts-of-speech and the same word length as the true word significantly more often than chance (Fig. 5C, $p < 0.005$). This result suggests that incorrect predictions capture syntactic as well as some perceptual features. For some of the datasets, we also observe that sublexical features such as the first or last letter of the word are better decoded than chance. Note that these features are typically well encoded in the word embeddings: a linear regression trained to predict word length from the word embedding of the 5000 most common English words yields a Pearson correlation of 0.71 ($p < 10^{-170}$). Similarly, a ridge classifier trained on these word embeddings can predict the first letter of the word with an accuracy of 26%(chance: 12% $p < 10^{-11}$). This phenomenon is expected, as e.g., in English, ending in 'ed' generally marks the past tense and starting with 'wh' generally marks a question.

The comparison of the datasets presenting the same sentences to participants, either through a reading or a listening task, suggests that decoding reading relies more on visual features than decoding listening (Fig. 5A, B). Specifically, for both the LittlePrince and Schoffelen datasets, the length of the failed predictions matches that of the true words more often in the reading than in the listening condition ($p < 0.001$), while the opposite holds when considering the part-of-speech ($p < 10^{-7}$). Given the retinotopic structure of visual presentations and the fixed font size and position, this result supports the idea that reading decoders also rely on visual responses.

**Relationship with word frequency.** To test whether our model can decode words absent from the training set, we evaluate the *zero-shot* decoding accuracy (Fig. 6A, B, yellow). This analysis is designed to test whether we the decoder simply memorizes the exact M/EEG patterns elicited by each word, or whether it learns its underlying semantic features. Zero-shot decoding accuracy is significantly above chance ($p < 0.005$), although its score is lower than for in-vocabulary words. This phenomenon appears to be linked to the fact that test words absent from the training set are typically rare in natural language, and hence harder to decode. Indeed, for in-vocabulary words, we observe that accuracy increases with the number of occurrences of the word in the training set (Fig. 6A).

**Relationship with part-of-speech.** Is decoding performance robust across different types of words? To address this question, we evaluate our decoder as a function of the part-of-speech categories (Fig. 6C, D) of the words considered. We observe a similar pattern for all datasets: performance is significantly above chance for all word lengths and part-of-speech categories, but is higher for function words ($p < 0.005$). This result is consistent with the relationship between decoding performance and word frequency, as function words are repeated many times in the training set. Overall, these results show that our decoder consistently decodes a variety of different words, but performs best with words repeated many times in the training set.

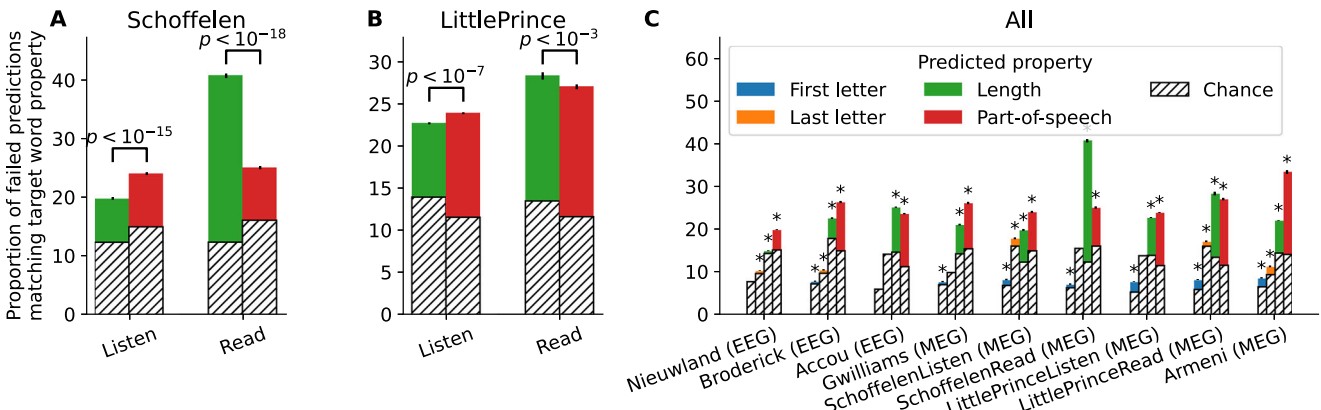

**Fig. 5 | Impact of sublexical and syntactic features on decoding.** For each incorrectly predicted word in the test set, we evaluate whether various properties of the top-1 prediction match those of the target word. **A**, **B** Results for length and part-of-speech, where the stimuli are identical between the listening and reading tasks. **C** Results for all properties (first and last letter, length and part-of-speech) across all datasets. Stars indicate significantly above chance classification ($p < 0.005$, one-sided $t$-test). In all panels, the error bars represent the SEM across subjects (the number of subjects for each dataset is reported in Table 1).

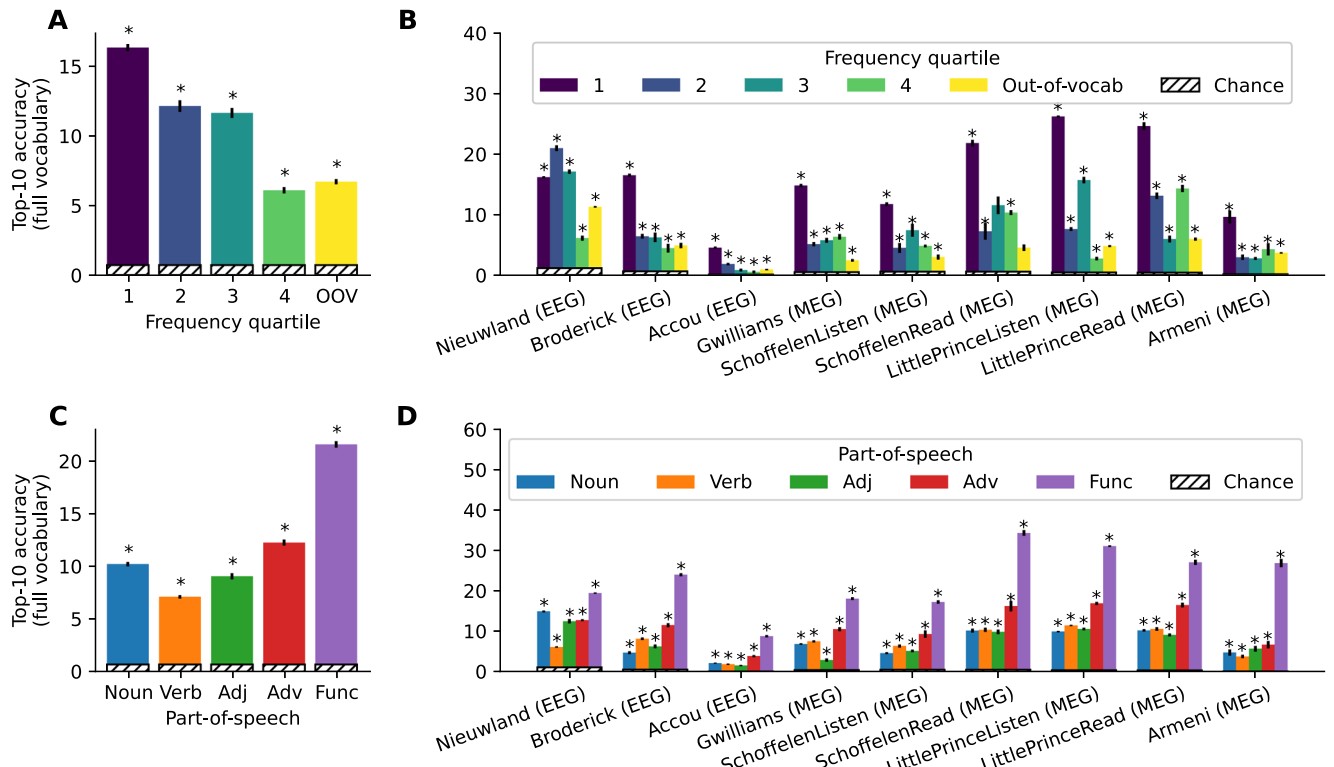

**Fig. 6 | Impact of various word properties on decoding performance.** **A**, **B** Impact of the number of occurrences of the word in the training set. We split the test set into quartiles according to the number of occurrences of each word in the training set, and denote as "out-of-vocabulary" words which do not occur in the training set. **C**, **D** Impact of the part-of-speech, obtained via Spacy. 'Function' contains all part-of-speech types other than nouns, verbs, adjectives and adverbs. In all panels, stars indicate significantly above chance classification ($p < 0.005$, one-sided t-test), and error bars represent the SEM across subjects (the number of subjects for each dataset is reported in Table 1).

## Discussion

In this study, we seek to evaluate the potential of a deep learning model to decode words from non-invasive recordings of a large number of participants and across a variety of recording devices, tasks and languages. For this, we curated an unprecedentedly large M/EEG dataset, encompassing 723 participants reading or listening to isolated or contextualized sentences, and used it to train and evaluate a new architecture optimized to decode the meaning of individual words.

Our work advances the development of BCIs by showing that a single AI architecture can decode single words from brain activity recorded with a variety of non-invasive recording devices, and under a variety of experimental conditions.

**Improving decoding performance from non-invasive recordings**
Technically, our decoding workflow outperforms classic methods such as linear models[19,30], EEGNet[22] and BrainModule[17] by a large margin. By

leveraging the power of transformers[31], our model operates at the sentence level and can exploit the surrounding context to improve the prediction of a given word.

Our approach also surpasses previous studies in two main ways critical to the development of real-time BCIs for natural language processing. First, and contrary to[16]'s fMRI study and subsequent works[32–36], our model aims to recover the semantics at the scale of *individual* words, rather than sentences. Second, and contrary to[17], our approach is based on word semantics, and does not require having access to the true spoken sounds in a retrieval set. Additionally, compared to the model used by[17], which operates on single words, our model operates at the sentence level thanks to the transformer, and is validated on a much larger panel of datasets. To the best of our knowledge, our model is the first to enable open-vocabulary decoding at the individual word level from non-invasive brain recordings.

Finally, our analyses highlight the impact of experimental designs. In particular, we show that MEG tends to be associated with better decoding performances than EEG. Similarly, the decoding of reading tends to be better than that of listening. Finally, decoding appears to be best when many recording hours are collected for each participant, rather than when many participants are recorded with the same stimuli for a short amount of time.

However, it is important to stress that the datasets presently analyzed differ across a wide variety of factors, such as the number of subjects, precise tasks, type of words and languages. Consequently, future work remains necessary to ensure that the comparison across devices or experimental conditions holds true under minimally varying experimental protocols. This effort, recently initiated in the domain of image perception[37,38], would require participants to be presented with the very same sentences across a variety of tasks and experimental apparatus.

The Little Prince MEG datasets presented here were acquired in both the reading and auditory modality to precisely contribute to a recent multi-site effort where participants from various countries are exposed to this book while their brain activity is recorded with various devices[39,40]. We hope that these datasets will contribute to clarify how differences across laboratories can be attributed to device, task and/or cognitive factors.

## How AI helps modeling the neural bases of word semantics

From a theoretical standpoint, our results confirms that the embeddings learnt by language models provide a remarkably useful tool to model semantic representations in the human brain[41–45]. This approach was originated by[18], who showed that isolated words could be decoded from fMRI with a simple linear classifier targeting a word embedding trained with a latent semantic analysis.[44,46] observed that various types of word embeddings, learnt for example from co-occurrence statistics of words in natural language, can reliably account for brain responses to natural stories. Since several studies systematically compared pre-trained language models to the fMRI[20,47–50], intracranial[19,51] and M/EEG responses to spoken[30,52] and written sentences[30]. Critically, several studies showed that this mapping between brain activity and word embeddings could be used to decode, in a zero-shot fashion, words out-of-vocabulary[18,19,30].

Here, we further show that M/EEG responses to words lead to decoding predictions that can be surprisingly close semantically. The similarity between word embeddings and brain signals suggests that there exist general principles to the organization of semantic features in biological and artificial neural networks, and that these principles transcend the precise architecture or training schemes of these neural networks. The present study illustrates the exciting potential of A.I. in elucidating how the human brain structures symbols and knowledge.

Our decoding analyses further suggest that non-semantic features – such as part-of-speech, word length and even single letters–also contribute to the decoders' predictions. These properties could both reflect sensory representations (*e.g.*, word frequency is correlated with the size of the word on retinotopic maps) or linguistic representations (*e.g.*, rare words may be more surprising). However, the complexity of our decoding pipeline fundamentally limits our ability to understand *how* semantic representations are articulated and disentangled from syntactic and sensory features. This remains an open challenge for both neuroscience and AI[53], for which encoding models may be better suited than decoding models[54].

## Path and challenges for building non-invasive brain-to-text decoders

Our decoding pipeline consistently scales with the amount of non-invasive brain recordings. However, there is a long way to go before this approach can be translated to practical applications. First, the present study focuses on language *perception* and not on language *production*. Second, single-trial performances remain far from those achieved with intracranial electrodes[5,55–57]. For example, given a vocabulary of 50 words, we reach a top-1 decoding accuracy of 20%, whereas[5] achieved a top-1 accuracy of 39.5% with an electrode implanted in the motor cortex. The limited performance of a non-invasive decoder seem to be mainly challenged by signal-to-noise ratio. Indeed, averaging over multiple M/EEG responses to the same words rapidly leads to high accuracies (Fig. 3B).

Several elements may partially improve the present decoding pipeline. First, we here focus on single-word decoding and thus require knowing the timing of word presentations. Other approaches typically used in speech transcription often use a Connectionist Temporal Classification (CTC) loss to circumvent this issue[58]. Second, while single-word decoding can be remarkable, the decoded sentences are often devoid of a clear meaning. This is expected, as a single word can suffice to break the grammatical structure of a sentence. To tackle this issue, language models input with the decoded predictions could improve the meaning of decoded sentences and narratives (*e.g.*[4,16]).

The biological bases of language, once an enigma, continue to yield its structure to the probing eyes of deep learning. This study indeed reveals the tantalizing prospect of decoding, at scale, the neural code of natural language, a feat that could not only democratize brain-computer-interfaces, but also expand our understanding of human cognition and its specificity in the animal kingdom.

# Methods
## Problem statement

The goal of the present study is to decode individual words from brain activity, including words absent from the training set. Such "zero-shot" decoding has already been shown to be statistically possible in[18,19,30] using linear models, but the corresponding performances are too low to enable decoding in practice.

Formally, we aim to predict pretrained word embeddings $Y \in \mathbb{R}^d$, from windows $X \in \mathbb{R}^{t \times n}$ of $t$ time-steps of brain activity recorded on $n$ sensors (note that this assumes knowledge of the word onset timings). Formally, we seek a mapping $f : \mathbb{R}^{t \times n} \rightarrow \mathbb{R}^d$ such that $\hat{Y} = f(X) \simeq Y$. This is illustrated in Fig. 1B.

## Objective

**CLIP.** Following[17], we rely on contrastive learning to map the brain responses to the word embedding space. Defining the cosine similarity as follows:

$$\measuredangle_{ij} = \frac{\hat{Y}_i \cdot Y_j}{\| \hat{Y}_j \| \| Y_j \|},\qquad(1)$$

Our objective is to maximize the cosine similarity for positive pairs $i = j$ and minimize that of negative pairs $i \neq j$.

The CLIP loss[59] treats this problem as a multiclass classification problem via the softmax function:

$$\mathcal{L}_{\text{CLIP}} = -\frac{1}{N}\sum_{i=1}^{N}\log\left(\frac{e^{t\measuredangle_{ii}}}{\sum_{j=1}^{N}e^{t\measuredangle_{ij}}}\right), \tag{2}$$

where $t$ is parameterized as $t = \exp(t\prime)$ with $t\prime$ a learnable parameter.

**D-SigLIP.** In our setting, there may be several repetitions of the same word in the batch, triggering different brain responses. This makes the contrastive learning problem ill-posed, as there would be matching (brain response and word embedding) pairs with a negative label.

This issue can be addressed with a SigLIP loss[60]. The SigLIP loss[60] was originally introduced to improve the scalability of language-image pretraining to large batches. It treats each element of the batch as $N$ binary classification problems, which dispenses with the need to compute normalization factors across the batch:

$$\mathcal{L}_{\text{SigLIP}} = \frac{1}{N}\sum_{i=1}^{N}\sum_{j=1}^{N}\log\left(\frac{1}{1+e^{z_{ij}(-t\measuredangle_{ij}+b)}}\right), \tag{3}$$

where $b$ is a learnable bias and $z_{ij}$ is usually equal to $+1$ for $i=j$ and $-1$ for $i\neq j$. In our case, we can also use $z_{ij}$ to define several positives for a given element of the batch. However, if we assign positive labels to all matching pairs, the fraction of positive to negative labels for a given word scales proportionally to the word frequency, which can cause class imbalance.

To alleviate this issue, in case of repeated words, we discard the repetitions from the loss: we call this modified loss D-SigLIP, for "Deduplicated SigLIP", see Supplementary Fig. 4 for more details and ablations.

## Models

**CNN.** To learn $f$, we first use the BrainModule model of[17]. It consists of (i) a spatial attention module combining the data from the different sensors given their spatial positions, (ii) a subject-dependent layer which handles inter-subject variability, and (iii) a stack of convolutional blocks.

To transform the dynamical output of the CNN, of size $t \times d$, to a static word-embedding $\hat{Y}$ of size $d$, we need to pool the temporal dimension. For this, we use a single-head self-attention layer with a unit output dimension.

**Transformer.** The CNN only processes a single window containing the target word, and is not input with the surrounding context. To better exploit context, we propose a new decoding architecture which consists in adding a transformer on top of the CNN, as illustrated in Fig. 1. Specifically, we split the text into sentences, each word of the sentence is fed to the CNN independently, and the outputs are stacked to form the input sequence of the transformer.

The transformer uses 16 layers and 16 attention heads, and its input dimension is equal to that of the target word embeddings, i.e., 1024. It uses an attention dropout of 0.1 and rotary positional embedding, following the implementation of the `x-transformer` package.

Note that this transformer is not a pretrained language model, and can thus be evaluated on its ability to retrieve information from brain signals, without leaking pre-existing linguistic knowledge.

**Baseline models.** We compare our deep learning pipeline to three baselines.

The first is a standard ridge regression model, implemented via SKLearn's `RidgeCV`[61], using a logarithmic search for the regularization parameter `alpha` varying from $10^{-2}$ to $10^{8}$. It is used to predict each component of the target word embedding separately (with a different regularization parameter for each component) from a single time point of M/EEG data. An independent model is trained for each time point between $-0.5$ and $2.5$ s, sampled at 50 Hz.

The second is EEGNet[22,62], a standard convolutional network used across the brain decoding literature. Finally, we compare the results obtained with our full pipeline with those obtained using only the BrainModule[17], without the transformer on top. EEGNet and Brain-Module are trained on all channels and all subjects of each study, in the same setting as our full pipeline.

## Evaluation

**Word retrieval.** The models described above predict word embeddings, rather than words directly. To obtain the corresponding words, we select from a fixed vocabulary $\mathcal{V}$, by identifying the word whose embedding has the highest cosine similarity to the predicted embedding $\hat{Y}$: $\text{argmax}_{Y\in\mathcal{V}}\measuredangle(\hat{Y}, Y)$.

The above word-prediction step depends on the chosen vocabulary, as demonstrated in Supplementary Fig. 5. However, vocabulary size varies substantially across datasets (see Table 1). Consequently, when comparing datasets, and unless stated otherwise, we report our metrics at a fixed vocabulary size by selecting for each dataset the 250 most frequent words. This reduced vocabulary covers between 70 and 95% of all word occurrences for the various datasets considered. Words falling outside of the reduced vocabulary are discarded from the evaluation metrics.

**Accuracy.** Top-1 accuracy is typically too strict to assess semantic decoding, as it would count as incorrect situations where our model predicts a synonym of the true word. Hence, we use **top-10 accuracy**, which measures how often the correct word is within the model's top 10 predictions.

Words in natural language are highly imbalanced: the ten most frequent words typically account for around 25% of word occurrences. Consequently, evaluation metrics may be biased by the most frequent words. Consequently, we compute a **balanced accuracy**, where accuracies are separately averaged per word, then averaged across words.

Unless specified otherwise, we will use this single-trial balanced top-10 accuracy on the 250 most frequent words throughout the paper, and will refer to it as "Top-10 accuracy" for convenience.

**Averaging over repetitions.** The above evaluation metrics focus on single-trial decoding, so as to provide a fair estimate of the decoding performance in a real-time setup. To further explore the extent to which our decoder captures a variety of linguistic features, such as semantics and part-of-speech, we also compute evaluation metrics of "averaged-trials". This approach improves the signal-to-noise ratio by averaging the predictions of the model across the multiple brain responses to the same word. We either average repetitions across different contexts for a given subject (*e.g.*, Subject 1: "To be or not **to** be"), or across different subjects for a given context (Subject 1: "To be or not to be"; Subject 2: "To be or not to be"), or both.

## Implementation details

**Word embeddings.** We obtained word embeddings by processing individual (non-contextualized) words with HuggingFace's `t5-large` model[63], and extracting the hidden representations from the middle layer. This design choice allows us to (1) use a single model across languages and (2) deal with multi-token words. Indeed, when a word is split into several tokens, we average the contextualized embeddings of the resulting tokens to obtain the target word embedding. Interestingly, as shown in Supplementary Fig. 3, the particular model used to obtain word embeddings has a mild effect on decoding performance.

**Training.** We train our model using the AdamW optimizer[64] using a learning rate of $10^{-4}$ and a batch size of 64. We use a cosine learning

rate decay over the first 50 epochs, and use early stopping based on the balanced top-10 accuracy on the validation set. Each run presented in this paper takes a few hours on an A100 GPU with 40GB of RAM.

## Datasets

To evaluate our ability to systematically decode individual words from non-invasive brain recordings, we consider datasets that provide a large amount of temporally-resolved brain responses to language. For this, we surveyed the main public databases, namely Osf.io, Datadryad, OpenNeuro and the Radboud University Data Repository. We identified 7 relevant EEG or MEG datasets, where the task either involved speech comprehension or reading through a "rapid serial visual presentation" (RSVP) protocol, where words are flashed at the center of the screen, one after the other. We add to this two new MEG datasets where participants either listened to or read "Le Petit Prince", by Antoine de Saint-Exupéry. All datasets were collected in accordance with participant consent and institutional ethics requirements.

These datasets not only vary in terms of brain recordings (number of participants, number of recordings per participant, type of recording device), but also in terms of language stimuli (language, reading vs listening, decontextualized sentences vs narratives), see Table 1. Importantly, we trained and tested the same architecture on each of these datasets separately. Training a single model jointly across multiple datasets did not lead to any performance boosts (see Supplementary Fig. 1). We speculate that the diversity of the datasets, which allows us to evaluate decoding performance across a variety of data regimes and experimental conditions, also hinders cross-dataset generalization. We leave the exploration of this challenge for future work, as it likely involves different training paradigms such as self-supervised pre-training[65–67].

**Nieuwland.** Nieuwland et al. [28]'s dataset is a preregistered EEG study involving nine distinct laboratories, which recorded a total of 334 participants (due to preprocessing issues, we discarded the recordings coming from the University of Stirling, leaving 295 participants for our study) while they read English sentences in an RSVP protocol. This study originally aimed to test whether the brain preactivates the phonological form of predictable words.

**Broderick.** Broderick et al. [68]'s dataset corresponds to the first EEG experiment of their study, where 19 subjects listened to the audiobook "The Old Man and the Sea". This study originally aimed to evaluate whether semantic features could be linearly retrieved from the EEG responses to words.

**Accou.** Accou et al. [27]'s dataset, *a.k.a* "single-speaker stories dataset," consists of 85 participants (the recordings of 5 of these participants are not publicly available) who listened to audiobooks and podcasts in Dutch while being recorded with an EEG. The goal of this study was to develop a deep neural network trained to decode the audio volume. Hence, the transcripts and word timestamps of the stimuli presented to the participants were not provided: we extract them using `WhisperX`[69].

**Armeni.** Armeni et al. [26]'s dataset is a MEG study involving three participants who listened to ten one-hour-long segments of "The Adventures of Sherlock Holmes". The goal of this study was to offer a dataset where a large amount of data is collected for each participant.

**Schoffelen.** Schoffelen et al. [70]'s dataset, *a.k.a* "Mother Of All Unification Studies", consists of 204 participants who either read or listened to isolated sentences in an MEG scanner. This study was originally designed to explore the brain responses to a variety of syntactic and compositional structures. The participants who read the sentences are disjoint from those who listened to them, but the sentences are shared between these two subsets.

**Gwilliams.** Gwilliams et al. [71] dataset, *a.k.a* "MEG-MASC" consists of 27 participants who listened to approximately 2 h of stories in the MEG scanner. Each story was repeated twice. The goal of this study was to provide a high-quality MEG dataset for encoding and decoding analyses.

***The Little Prince* datasets.** Using a 306 MEG Elekta Neuromag machine, we collected two MEG datasets in which 102 native French healthy volunteers were presented with the full story of *Le Petit Prince* by Antoine de Saint-Exupéry (see[39] for a detailed description of the stimuli). 46 participants read the text, while 58 participants listened to it.

In the case of reading, an RSVP paradigm was used to avoid eye movements. Words were flashed at the center of the screen every 300 ms—each word was displayed for 250 ms followed by a 50 ms blank screen − and sentences were separated by an additional blank screen lasting 500 ms.

Aside from the perceptual modality, the main differences between the two datasets are the following: (i) LittlePrinceListen has a larger total duration because it has more subjects but also because of the slower delivery of audio speech (100 min vs. 90 min for the visual presentation), (ii) the number of tokens per subject and vocabulary size differ because of the difference in segmentation of words (for example, "j'avais" is segmented as ["j'", "avais"] for the listening dataset and as ["j'avais"] in the reading dataset).

These datasets were acquired at the Neurospin Center, CEA, Gif-sur-Yvette, by authors CB and CP. The protocol (CEA 100 049 / ID RCB: 2018-A02586-49) was reviewed and approved by the Comité de Protection de Personnes Sud-Est VI Clermont-Ferrand (ethics committee).

**Preprocessing.** Given the diversity of datasets considered, preprocessing was kept to a small common denominator: the M/EEG recordings were bandpass filtered between [0.1,40] Hz and resampled to 50 Hz, using built-in functions from `MNE-Python`[72], then scaled using `sklearn`'s `RobustScaler` and clamped in the range [−5, 5]. We do not perform any artifact or bad channel removal in this work, relying on the deep learning model to discard undesirable features.

**Splitting.** We split the recordings into 3 s windows, where each window starts at the word onset, and baseline correction is applied to the neural data over the first 0.5 s. Note that while the baseline is usually computed on a segment of the data occurring before the stimulus, in our case, this did not change the decoding accuracies achieved, and hence, we reuse data from the stimulus window for efficiency (see Supplementary Fig. 2).

We split the train, validation and test data with an 80/10/10 ratio. To avoid the data leakage observed in many language decoding papers[73], we ensure that the same sentences presented to different subjects are assigned to the same split by hashing them deterministically.

## Inclusion and ethics

A total of 104 native French speakers participated in this study. All participants were right-handed, had normal or corrected-to-normal vision (no color blindness) in the reading experiment, normal hearing for the listening condition, and reported no history of neurological disorders. Participants were divided into two groups: 46 (7 females, mean age = 28.6, SD = 5.7) read the text of Le Petit Prince, and 58 (17 females, mean age = 27.4, SD = 5.3) listened to the audiobook. The study protocol (CEA 100 049 / ID RCB: 2018-A02586-49) was reviewed and approved by the Comité de Protection de Personnes Sud-Est VI Clermont-Ferrand (ethics committee). All participants provided

written informed consent prior to participation. Following informed consent, participants were familiarized with the MEG environment. For the listening group, participants were comfortably seated and instructed to minimize movement. Once seated comfortably in the MEG seat, participants chose their ideal stimulus volume by determining a level that was loud and comfortable. Auditory stimuli were presented via headphones, fitted to their inner-ear shape. Participants were instructed to listen attentively to the Le Petit Prince audiobook. For the reading group, participants were seated in front of a screen and instructed to fixate on the center. The text of Le Petit Prince was presented using a rapid serial visual presentation (RSVP) paradigm, with words displayed briefly at the center of the screen to minimize eye movements. The entire MEG recording session lasted approximately 90 min for the reading group and 100 min for the listening group. It was for both conditions, divided into 9 runs of approximately 7–10 min each. They also completed 4 comprehension quiz questions after each run (36 questions in total). These questions, designed to confirm their understanding, were presented on the stimulus screen, and participants responded using a button box. During the MEG recordings, participants were monitored via a camera to ensure they remained alert and still. If participants appeared drowsy or exhibited excessive movement, the operator provided a warning once the run was finished. During the breaks between runs, participants were instructed to relax but remain stationary. After completing the session, participants were compensated and sent home. The total session time, including preparation and recording, lasted approximately 2.5 h. Participants are compensated 80 euros per session.

### Reporting summary

Further information on research design is available in the Nature Portfolio Reporting Summary linked to this article.

## Data availability

Due to legal constraints, we will share data to academic researchers upon request. Requests should be send to christophe.pallier@cea.fr.

## Code availability

The code used in this paper is provided as a zip file in the supplementary material.

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

## Acknowledgments

The authors thank Fosca Al Roumi, Leila Azizi and Florent Meyniel for their support on the experimental protocol, Robin Schirrmeister and Loïc Barrault for useful discussions, and Pierre Louis Xech and Abhishek Charnalia for program management.

## Author contributions

S.D. and J.D. conceived and designed the study. S.D. performed the experiments and carried out the computational analyses. S.D., J.R., H.B., Y.B. and J.K. contributed to the codebase. The experiments in this paper were designed by SdA and JK and performed by SdA. C.B. and C.P. conceived the protocol of the LittlePrince 2wazand C.B. performed the data collection. S.D. wrote the manuscript with input from all authors.

## Competing interests

The Authors declare no competing interests.

## Additional information

**Peer review information** : *Nature Communications* thanks Elvira Khachatryan, Cheng Luo and the other anonymous reviewer(s) for their contribution to the peer review of this work.

