## [Transparent Peer Review file · Nature Communications]

Towards decoding individual words from non-invasive brain recordings

Corresponding Author: Dr Stéphane D'Ascoli

Version 0:

Reviewer comments:

Reviewer #1

(Remarks to the Author)

Summary:

This manuscript presents a comprehensive study in the field of brain-computer interfaces (BCIs), aiming to decode individual words from non-invasive electroencephalography (EEG) and magnetoencephalography (MEG) signals using a deep learning pipeline. The authors collected a large MEG/EEG dataset involving 723 participants engaged in reading or listening to isolated or contextualized sentences across three languages. Utilizing this extensive dataset, the authors evaluated their deep learning pipeline across different recording devices (MEG and EEG), tasks (reading and listening), and languages (English, French, and Dutch). Their approaches consistently outperform existing methods and improve “zero-shot” decoding performance. The key findings include: 1) MEG recordings during reading tasks are more readily decodable than EEG recordings during listening tasks; 2) Decoding performance improves with larger data used for training and per-word averaging; 3) Collecting extensive data per participant yields better decoding performance than increasing the number of participants with repeated stimuli; 4) The decoding model can capture both semantic information and syntactic/surface-level features in single-word predictions.

Overall, this is a commendable study that addresses a long-standing challenge in BCI research through the use of a substantial dataset, refined methodologies, and meticulously executed analyses. The writing is clear and accessible, and the findings are largely consistent with expectations. However, several major concerns and a few minor issues require further consideration.

Major:

1. In the Methods section, the authors have chosen to utilize the T5-large language model to generate word embeddings for different languages, which is a reasonable choice (lines 317-324). However, it is well-known that the choice of language models can affect the quality and characteristics of word embeddings, which in turn may influence the word decoding performance. To further strengthen the manuscript, I would suggest that the authors consider discussing any potential limitations or biases that may arise from using the T5-large model. It would be valuable to explore how these factors might impact the decoding results.

2. The authors mention that they split the recordings into 3-second windows, with each window starting at word onset, and that baseline correction is applied to the neural data over the first 0.5 seconds of each window (lines 308-310). I would like to suggest a clarification regarding the baseline correction procedure. Typically, baseline correction involves subtracting the average signal from a period before the word onset, rather than from the initial part of the window following the word onset. Therefore, it would be more appropriate to apply baseline correction using the 0.5-second period preceding the word onset. This ensures that any neural activity directly related to the word processing is not inadvertently removed or altered by the baseline correction. Please confirm whether this adjustment was made or provide additional justification for the chosen method. Clarifying this point will help ensure the accuracy and interpretability of the results.

3. The current study presents a deep learning pipeline based on MEG/EEG signals from different datasets, encompassing both reading and speech comprehension tasks. The results indicate that the model exhibits better decoding performance for MEG signals / reading tasks compared to EEG signals / speech comprehension tasks (Fig. 2D). However, it is not explicitly

stated whether the authors trained separate models on each dataset individually or trained a single model on all datasets combined. If the authors trained separate model on each dataset, it would be important to clarify how consistency was maintained across datasets in terms of data scale, signal quality, and other relevant factors. Ensuring such consistency is critical for avoiding discrepancies in decoding results attributable to variations in data quality or quantity. If, on the other hand, all datasets were combined to train a single model, followed by separate evaluation on the test sets of each dataset, it should be noted that neural activity patterns can differ significantly between speech comprehension and reading tasks. This raises the possibility that the model might have been more optimized for capturing the characteristics of brain activity associated with reading tasks rather than speech comprehension tasks. This could potentially skew the decoding performance in favor of reading tasks. I would like to ask whether the authors have considered training word decoding models separately for reading tasks and speech comprehension tasks. Comparing the performance of these task-specific models against the current unified model would provide valuable insights into any potential biases introduced by the training procedure based on the mixed-task signals.

4. The deep learning pipeline demonstrates commendable decoding performance across various datasets, with the overall decoding performance surpassing that of three baseline models: Linear, EEGNet, and BrainModule. However, I notice that for certain datasets, such as Nieuwland and LittlePrinceRead, the current model exhibits inferior performance relative to BrainModule (Fig. 2B). Could the authors explain why this discrepancy occurs? Specifically, exploring the reasons behind the reduced effectiveness of the proposed model on certain datasets would be beneficial. This could provide valuable insights into the strengths and limitations of the current approach and highlight areas for further improvement.

5. The current manuscript shows that incorrect predictions share the same word length as the true word significantly more often than would be expected by chance (Fig. 5). Based on this observation, the authors argue that their decoding model captures visual features, specifically word length, during the training process (e.g., lines 466-469, and 492-502). I find this claim somewhat confusing. The current deep learning pipeline utilizes word embeddings generated by the t5-large model, which primarily focus on the semantic meaning and contextual relationships of words rather than their physical length. Moreover, word embeddings do not inherently encode any visual features of the words themselves. Given that word embeddings serve as the output target in the current pipeline, it is unclear why the model would capture visual features in the training process. This point warrants further discussion and clarification from the authors.

Minor:

1. The current study uses top-10 accuracy, which measures how often the correct word is within the model's top 10 predictions. From a practical standpoint, this metric may be too lenient. Specifically, in real-world applications, users typically expect the correct prediction to be among the top few choices rather than being buried within a broader set of 10 options. Therefore, it is worth considering whether the top-10 accuracy provides an overly generous assessment of the model's performance. A more stringent evaluation, such as top-3 or top-5 accuracy, might better reflect the model's practical utility. Additionally, discussing the implications of using top-10 accuracy for the model's real-world applicability would be beneficial.

2. It appears that some of the line plots in Fig. 3A are missing the shaded regions indicating standard deviation.

3. Some typos need to be corrected.

e.g., Table 2: "...Bold *indicate (->indicates) incorrect predictions..."

Line 208: "for read (->reading)."

Line 212: "...our decoder capture (->captures) a ..."

Line 248: "...the brain preactivate (->preactivates) the phonological..."

Reviewer #2

(Remarks to the Author)

This study by Ascoli et al. proposes a deep learning pipeline to decode individual words from non-invasive EEG and MEG signals. They train and evaluate their pipeline with a large number of 723 participants and included different languages. Their analyses showed higher decoding performance compared to existing methods. This manuscript is straightforward in its design so the logic is easy to follow, and presented clearly both in the written and graphic elements. However, a number of points need clarifying and certain statements require further justification. My detailed comments are as follows:

The proposed methodology lacks a compelling demonstration of its novelty, as the authors did not explicitly delineate the distinctive features that differentiate their deep learning pipeline from existing approaches, particularly in relation to the model presented in the referenced prior study. A more rigorous comparative analysis is essential to substantiate the claim of a genuinely innovative contribution, rather than presenting what appears to be an incremental modification of established techniques. Regarding model selection, the rationale behind choosing the Transformer architecture remains unclear. The authors should provide a comprehensive theoretical justification that elucidates the architectural design choices.

I am also wondering whether the authors have tried directly encoding neural signals from MEG data as a neural signal encoder, then generating a text encoder directly for words, and subsequently using the CLIP model. Could the authors elaborate on the specific advantages, computational strategies, and theoretical rationale behind mapping raw brain signals into a standardized word embedding space, particularly in terms of potential improvements in neural decoding accuracy, semantic representation fidelity, and cross-modal translation capabilities?

The scale of the model appears to be carefully calibrated, balancing computational complexity with potential performance gains, though the authors could potentially elaborate on the specific trade-offs considered during architectural design. I am also wondering why the repetition of the same word within a batch evoked different brain responses. Could it be due to

semantic saturation or the dynamic allocation of attentional resources? Is a Multilingual Language Model trained and fine-tuned separately on each dataset? Are the training strategies consistent across different datasets? Have any adjustments been made to accommodate the specific characteristics of each dataset? Details should be added to address the above questions. Moreover, to comprehensively assess the proposed pipeline's effectiveness, it will be beneficial to conduct other ablation studies besides Dedeuplicated SigLIP.

The authors did not explicitly elaborate on the semantic features extracted by the proposed model or provide a comprehensive mechanism for revealing these semantic characteristics. Further investigation is needed to elucidate the precise semantic representations and feature extraction processes underlying the model's decoding strategy. The results show that the decoding performance is highly dependent on extensive individual-level data collection, which potentially indicates limitations in the model's ability to generalize neural representations across different participants, raising questions about the model's robustness and transferability of neural encoding patterns. These issues should be thoroughly addressed in this work.

Reviewer #3

(Remarks to the Author)

The authors developed a new decoder by adding a layer of transformer to CNN and attempted to decode the words from the brain signals using different publicly available databases as well as their own MEG recordings. The topic is now on a rise and attracts a lot of attention considering the potential use in BCI.

First of all, even though combining more than 700 subject recordings sounds very impressive, the abstract and their claim of developing the dataset is very misleading considering that they use mostly already available data and they didn't even combine the datasets. They train their model on each dataset separately and considering that the model is a black box it can adjust its parameters on every dataset.

The other claim that it is better to have many recordings from few subjects than visa versa is wrong. In that case the model will indeed perform better (unsurprisingly) on that dataset because intrasubject variability is always smaller than intersubject variability. But then one runs into the risk of overfitting the model. In that case the model and the obtained results will not be possible to extrapolate on a larger population.

One should also be very careful with the claim that the model bases itself on semantics because the interaction between length of the words, the frequency of word-occurrence and the letters found in those words was not properly explored. The fact that read words were better decoded than heard ones was not surprising considering that the former ones are processed as one also based on their visual features, while the latter processed gradually as an auditory information arrives

Additionally you can find below some specific comments on the text:

Page 1, line 73: what is the performance accuracy?

Page 5, line 308: did the baselining occur with the part of the trail that contains response to stimulus? As each window began with a word. It was 3s the duration of a sentence? That is not clear

Page 6, line 343: is statistics corrected for multiple comparisons?

Page 7, lines 371-372: type? Even though it doesn't feel like the accuracy difference would be significant

Page 7, line 383-384: I don't understand this. Exposing subjects to the same sentence and comparing reading with listening is not the same

Page 7, lines 392-394: then the results of the experiment cannot really be extrapolated to other individuals. Aren't we then running into the risk of overfitting the model? Did the author look at how the model did when the databases are combined? Given of course that the recording device (MEG or EEG) and the language are the same

Page 8, line 432-437: what do the authors mean by "context". In the methods section it seemed like they were averaging the words from the same sentence presented several times to the same subject. If it is so, the context is not different. Please clarify.

Page 9, line 503: what is the accuracy then? The figure is not very clear as the bar is too small. I don't understand the reasoning behind the statement that words outside the vocabulary are too rare. Why can't some frequent words be used in training and others in testing?

Page 11, line 554-557: I don't think this claim can be done considering that there was no comparison with retrieval of non-words. Besides considering that the functional words has the highest accuracy this claim is even more doubtful

Besides the clarification of above mentioned specific comments, I could give the following suggestions to improve the article:

1. tone the conclusions and the abstract down considering that the dataset is not collected by the authors and the datasets from different papers are not merged
2. I would try to train the model on one dataset and test on the other (given some words overlap in different datasets). If it performs better than other models, the added value of this model then will be shown.
3. I missed in the methods the information on how much data was used for training and how much for testing. Was there an overlap between training and testing set?

4. The accuracies should be mentioned in the text considering the difference on the figures is not that clear.
5. I would also discuss the the question whether the shown difference is relevant despite it being statistically significant, considering how small it is

Taking into account the above mentioned comments, my recommendation for this article in the current condition would be to not consider it for the publication.

Reviewer #4

(Remarks to the Author)

The authors propose and evaluate a novel method to decode words from human electrophysiology data during language comprehension (total of 9 datasets, EEG or MEG). This was an impressive and computationally challenging undertaking, testing the robustness and scalability of language decoding from non-invasive brain recordings across 9 datasets of varying lengths and sizes, 3 indo-European languages, 2 task modalities (reading and listening), and 2 recording modalities (EEG and MEG).

This report contributes significantly to development and validation of methods in decoding non-invasively recorded brain signals. The improvements of their proposed pipeline, over reasonable baseline methods, are considerable (e.g. up to ~50% boost with their pipeline, translating to ~8% increase in accuracy, on average across datasets). Beyond accuracy boosts, this report thoroughly investigates the various factors and design decisions that influence brain decoding pipelines with the same experimental setup (number of words to decode from, dataset size, reading vs. writing). Typically studies will focus on one task or language at a time (e.g. just reading, English-only, just EEG etc.), so it's very valuable to see the evaluation of several datasets with common pipeline(s). It's an impressive effort in data curation and a good addition to the current literature.

I have no problems seeing this report published, but I do have a small number of mostly clarification points and editing suggestions that I would like to see addressed before I can recommend it for publication. I hope the authors find my comments sufficiently clear and useful in revising the manuscript. I believe the authors should not have too much trouble addressing my concerns.

Major

1. Preprocessing. The preprocessing section is quite short. I understand that given the extensive data curation, you kept the preprocessing steps to a smallest common denominator. I do wonder, however: were you concerned with data quality within datasets (artifact and bad channel removal) at all? Or were all datasets already preprocessed? Or did you simply fit to all channels and accept that bad channels would be dropped via poor decoding? If you had any heuristics in dealing with these issues, perhaps those would be useful to briefly document and briefly in an appendix or so (for readers that might attempt similar and are interested in pipeline development)?

2. Deduplicated SigLIP. The section on your novel deduplication loss function was not entirely clear to me. You motivate the choice to use SigLIP loss in order to be able to deal with repeated words within a batch (which CLIP is not designed for). Then you state that for Deduplicated SigLIP you "discard the repeated words". Meaning you don't train the model on repetitions, always just one occurrence of the word in the batch -- does that mean that d-SigLIP is (in terms of loss signals used to train the classifier) effectively the same as CLIP (Based on the Fig. 8 you arrive at the schema for CLIP again?). Given that d-SigLIP gives you a decoding boost (table with Fig. 8), I understand it is a different loss, but I fail to gather from your description what distinguishes it from CLIP.

Also: How do you choose which elements of the batch are discarded? Do you just retain the first element of the batch and drop the rest or is it done some other way? Does your dropping criterion bias your classifier in any way (e.g. if you keep only the first element in the batch, would that mean that the model learns from brain signals at sequence onset)? In my opinion, it would be good to report that for completeness.

3. Baseline regression model time-point selection. On l. 170 you state: "We vary the offset of the time-point relative to the word onset between -0.5 and 2.5 seconds." On my first pass, I thought you were randomly selecting time-samples, but based on Fig 2A, I think it simply means you trained independent models per each sample point? I'd try to clarify that.

4. Description of baseline models. I think it'd be helpful to add similar notation (dimensions of input features etc.) for baseline model description too, just like you do for the main pipeline. For example, are baseline regression models fit independently per M/EEG channel? Or were they fed all channels at the same time for each sample point? I would also move the more compact description of the baseline model from the results section (l 336-339) up the methods section (around line 165).

5. Fig 2.G. On l. 387 you state: "This suggests that with a fixed recording budget, it is better to record a small number of participants across many sessions than a large number of participants with a small amount of sessions." This is an interesting analysis. Admittedly, it looks like the linear trend ($R = 0.7$) is likely driven by one high leverage point; perhaps that warrants qualification? In addition, note that this conclusion applies if one is solely concerned with accuracy. If one has a different goal in mind (e.g. generalization across subjects, say, developing a single decoder that works across subjects) that might change the consideration. Given that you are highlighting this result in the abstract, perhaps you can add a cursory

comment about that too? Or do you think differently?

6. Fig 5A. Although, the analysis description is clear, I find parsing the figure somewhat difficult. What is the y-axis showing? How should it be read? Is it showing the % accuracy of for the target property? For example in Fig 5A, third bar (Schoffelen, green): does 40% accuracy mean that out of all test samples which are wrong (based on top-1 criterion), 40% of the time the decoded word will be the same length than the true word? Then, what do the hashed bars indicate?

In addition, the label "Accuracy (failed predictions)" was not entirely intuitive to me. How can there be accuracy if a prediction is wrong? I was able to understand your textual recap of the results just fine, but looking at the figure I got confused and captions were not helpful. Perhaps I'm missing something obvious, but it would be helpful to spell out in captions what the bar hashing indicates, how to read the y-axis etc.

7. Discussion. On l. 549. "This improvement is important to go beyond statistical metrics like the Pearson". It might just be a matter of wording, but why would a quantitative improvement in accuracy based on a new architecture speak to what metric one uses to test the model? If I read your work correctly, the message is that it is important to go beyond standard linear decoders (e.g. improving the loss objective, adding transformer modules for contextual effects etc.) rather than test metrics. Also, I don't see what is "statistical" about the Pearson correlation coefficients (compared to any other possible metrics). Might be helpful to clarify this statement a bit.

8. Discussion (l. 543). "Theoretically, it offers insights to the neural underpinning of language representations 545." I appreciate the attempt to add a theoretical angle to the discussion! But I do find this statement a bit of a stretch and the theoretical contributions here minimal (by necessity, given the strong methodological and engineering contributions, which I find totally fine). I agree that the work contributes to the long line of work using distributional models of word semantics and the cursory discussion of relevant past work is welcome. But distributional semantics is not tied to AI (i.e. artificial neural networks) as your discussion might suggest and you don't investigate competing lexical semantic approaches. Under "neural bases", I would expect a discussion/exploration of the role of different aspects of neural dynamics (spatio-temporal aspects, spectral power/phase etc.) in decoding which is beyond the scope of your work. I personally prefer concise statements of contributions (e.g. "Our work further shows the usefulness of distributional models of word semantics" ...). This just a suggestion.

Minor

1. Pooling across temporal dimension in CCN. You mention that you pooled the CNN representation across time using a single-head self attention layer. This is not clear in the diagram in Fig. 1B. For clarity, you could consider adding it explicitly, after the CCN layer and before transformer in the diagram too.

2. Manuscript layout. I'd personally move paragraphs in section 2.6 ("Implementation details") under section 2.4. ("Evaluation"). And perhaps you could rename section 2.4 to something like "Training and Evaluation".

3. Fig 2, caption A. "Decoding is evaluated with the average Pearson correlation on the test set." I think for readers this won't be clear. Perhaps saying something like "Predictive accuracy was evaluated as Pearson correlation across time per embedding dimension" or something similar. You could also formally describe this evaluation metric in the methods section 2.4 (and make clear it was only used for baseline models).

4. Fig 3C. Instead of using line style for differentiating two conditions (averaging over context vs. subjects) you could also consider using a different marker (e.g. circles vs. triangles etc.). I think it would just be a little more easier to distinguish.

5. Table 2. You should probably clarify a little bit what dataset/model etc. this decoding comes from (e.g. Example decoding for the best performing model/pipeline on dataset XYZ)

6. Discussion (l. 625). "Use a a Connectionist Temporal Classification (CTC) loss to circumvent this issue." Consider adding a reference to this method and/or references using this method.

Version 1:

Reviewer comments:

Reviewer #1

(Remarks to the Author)

All my concerns have been thoroughly addressed. The detailed responses provided, alongside the corresponding revisions to the manuscript, significantly enhance the clarity, rigor, and completeness of the work.

Cheng Luo

(Remarks on code availability)

Although I could not run the codes due to the lack of data, the authors have provided well-structured codes accompanied by detailed README files. I believe that the current results presented in the paper can, in principle, be reproduced using these

codes.

Reviewer #2

(Remarks to the Author)

The authors' revisions have significantly strengthened the manuscript through detailed clarifications, enhanced methodological transparency, and additional cross-dataset validation experiments, which collectively reinforce the rigor and generalizability of the findings. The inclusion of FastText embeddings, baseline protocol comparisons, and discussions on dataset heterogeneity further underscores the robustness of the proposed approach.

However, a few issues still require further consideration to fully solidify the study's contributions:

1. While the authors emphasized differences from prior research (Défossez et al., 2023) (e.g., task objectives, contrastive loss functions, sentence-level Transformers), how did they quantify the practical impact of these differences? How to understand the quantitateness represented by Figure 1B?
2. The authors attribute improved accuracy to "sentence-level operations" but omit critical details: Why is a Transformer more suitable for EEG/MEG time-series data than alternatives (e.g., CNNs, LSTMs)? How does self-attention specifically enhance alignment between neural signals and word embeddings? No analysis of attention patterns (e.g., visualizing which brain signal segments or frequency bands are prioritized) was provided.
3. The authors attributed differences in brain responses to "different subjects or contexts" but provided no empirical evidence (e.g., neural signal visualizations, attention weight distributions, or correlation analyses). To strengthen this claim, the following should be added: a) Quantitative analysis of neural signal similarity across repeated words (e.g., Pearson correlation coefficients); b) Contextual effects on decoding performance (e.g., sentence position, semantic relatedness). Additionally, does the authors' explanation implicitly assume that the observed variability in neural responses is unrelated to potential mechanisms such as "semantic saturation" or "dynamic attentional allocation"?
4. The authors state that T5 embeddings were used uniformly across datasets without fine-tuning, but some questions remain:
How were cross-lingual discrepancies in neural signal distributions (e.g., syntax, phonology) handled? Were language-specific biases in pretrained T5 embeddings (e.g., English-centric training) mitigated?

(Remarks on code availability)

The authors have provided code along with a README file. However, the current code is relatively simple and cannot be run without sample data. For better reproducibility and completeness, it would be helpful if the full code, along with (part of) the raw data, could also be shared.

Reviewer #3

(Remarks to the Author)

I would like to thank authors for the extensive work they did on revising this article. My concerns are mostly addressed. I would only like to emphasise that instead of saying that "(...) with fixed recording budget..." They should mention that choice for recording more data from less subjects should be guided by research question. For example, this approach might be more beneficial for bci studies where per subject performance improves with recording more data. However, as authors also mentioned, it will limit the interpretation of other goals as understanding the brain.

(Remarks on code availability)

Reviewer #4

(Remarks to the Author)

I thank the authors for their response. They addressed all my concerns.

(Remarks on code availability)

Thanks for including the code with submission! I have reviewed the submitted code briefly, but have not attempted re-executing, nor reproducing it. I can confirm that it contains the following:

- a license
- a list of dependencies and Python packaging information
- cursory documentation (e.g. README.md) and examples on how to setup the code and run experiments
- code tests

I am not sure how straightforward it would be to immediately re-execute the code. For example, just quickly looking at ``python -m sentence_decoding.grids.final`` from provided documentation, it seems like it would run over one dataset only as the ``grid`` variable has other dataset names commented out. However, it is a complex, sophisticated code base and in better shape than most research code.

I assume the authors will share it in a format (GitHub, PyPI or similar) that will make it easier for users to benefit from it, fix bugs, and possibly reproduce experiments.

Version 2:

Reviewer comments:

Reviewer #2

(Remarks to the Author)

The authors have addressed all my concerns.

(Remarks on code availability)

-

Rebuttal: Decoding words from noninvasive brain recordings

We thank the reviewers for their thorough and constructive reviews. First, we are happy to see that this study was judged as

- “impressive and computationally challenging undertaking [which] contributes significantly to development and validation of methods in decoding non-invasively recorded brain signals” [R4],
- a “commendable study that addresses a long-standing challenge in BCI research” [R1],
- “presented clearly both in the written and graphic elements” and that the “analyses showed higher decoding performance compared to existing methods” [R2].

However, the reviewers highlighted several major elements:

1. We only explored a single word embedding model (T5) [R1 and R2],
2. Our model was not trained jointly across datasets [R1 and R3],
3. Our baselining procedure is non-standard [R1 and R3].

To address these issues, we followed reviewers’ suggestions, and performed the following experiments:

1. We replicated our experiments with FastText, a seminal character-level word embedding. Our results are consistent across the two word embeddings, although T5 remains slightly better.
2. We conducted new experiments using a single model trained on multiple datasets, and our findings indicate that while multi-dataset training is effective, it underperforms compared to within-dataset training. We attribute this performance gap to the limited and heterogeneous nature of the current dataset collection in our discussion section.
3. We replicated our findings with a standard baselining protocol.

Overall, these additional experiments strengthen our original findings.

Finally, we would like to comment on a concern shared across reviewers: our model “relies on extensive individual-level data collection, which potentially indicates limitations in the model's ability to generalize neural representations across different participants” [R1]. We agree with this remark. However, we would like to emphasize that zero-shot generalization to new subjects is not the goal of this study. Achieving within-subject decoding is already a major challenge [1], especially for language tasks [2,3,4,5]. In addition, while our model cannot generalize to completely unseen subjects, decoding accuracy increases with the number of participants used for training (figure 3A). This result thus indicates that the model learns representations that are at least partially shared across participants, and hence contribute to a growing effort on improving the decoding performance obtained from a limited amount of single-subject data thanks to multi-subject training [6,7]. We now clarify this issue in the discussion.

Before addressing each of our reviewers' points below, we would like to, once again, thank them for their valuable remarks that helped improve our manuscript.

[1] Chevallier, Sylvain, et al. "The largest EEG-based BCI reproducibility study for open science: the MOABB benchmark." *arXiv preprint arXiv:2404.15319* (2024).

[2] Moses, David A., et al. "Neuroprosthesis for decoding speech in a paralyzed person with anarthria." *New England Journal of Medicine* 385.3 (2021): 217-227.

[3] Willett, Francis R., et al. "A high-performance speech neuroprosthesis." *Nature* 620.7976 (2023): 1031-1036.

[4] Metzger, Sean L., et al. "A high-performance neuroprosthesis for speech decoding and avatar control." *Nature* 620.7976 (2023): 1037-1046.

[5] Tang, Jerry, et al. "Semantic reconstruction of continuous language from non-invasive brain recordings." *Nature Neuroscience* 26.5 (2023): 858-866.

[6] Scotti, Paul S., et al. "Mindeye2: Shared-subject models enable fmri-to-image with 1 hour of data." *arXiv preprint arXiv:2403.11207* (2024).

[7] Tang, Jerry, and Alexander G. Huth. "Semantic language decoding across participants and stimulus modalities." *Current Biology* (2025).

Reviewer #1

Summary:

This manuscript presents a comprehensive study in the field of brain-computer interfaces (BCIs), aiming to decode individual words from non-invasive electroencephalography (EEG) and magnetoencephalography (MEG) signals using a deep learning pipeline. The authors collected a large MEG/EEG dataset involving 723 participants engaged in reading or listening to isolated or contextualized sentences across three languages. Utilizing this extensive dataset, the authors evaluated their deep learning pipeline across different recording devices (MEG and EEG), tasks (reading and listening), and languages (English, French, and Dutch). Their approaches consistently outperform existing methods and improve “zero-shot” decoding performance. The key findings include: 1) MEG recordings during reading tasks are more readily decodable than EEG recordings during listening tasks; 2) Decoding performance improves with larger data used for training and per-word averaging; 3) Collecting extensive data per participant yields better decoding performance than increasing the number of participants with repeated stimuli; 4) The decoding model can capture both semantic information and syntactic/surface-level features in single-word predictions.

Overall, this is a commendable study that addresses a long-standing challenge in BCI research through the use of a substantial dataset, refined methodologies, and meticulously executed analyses. The writing is clear and accessible, and the findings are largely consistent with expectations. However, several major concerns and a few minor issues require further consideration.

Major:

1. In the Methods section, the authors have chosen to utilize the T5-large language model to generate word embeddings for different languages, which is a reasonable choice (lines 317-324). However, it is well-known that the choice of language models can affect the quality and characteristics of word embeddings, which in turn may influence the word decoding performance. To further strengthen the manuscript, I would suggest that the authors consider discussing any potential limitations or biases that may arise from using the T5-large model. It would be valuable to explore how these factors might impact the decoding results.

Thanks for raising this important question. We have now re-run our experiments with word embeddings from FastText [1]. As described in the Supplementary Material,

"[these word embeddings] are constructed by representing each word as a bag of character n-grams, which are then used to learn a weighted sum of these character sequences. The model is trained on a large corpus of text using a variant of the skip-gram model, with the goal of predicting a target word given its context words. FastText incorporates subword information by combining character n-grams and the word itself, allowing it to handle out-of-vocabulary words by generating embeddings from their character n-grams."

In the figure below, we show that the average decoding performance across datasets displays a moderate drop from 23% to 20% top-10 accuracy when switching from T5 embedding to FastText embeddings. This result suggests that the choice of word embedding model is not critical when the goal is to decode non-contextualized word embeddings.

We report this result in the Supplementary Material, and state this in section 2.5:

"Interestingly, the particular model used to obtain word embeddings has a mild effect on decoding performance."

[1] Bojanowski, Piotr, et al. "Enriching word vectors with subword information. CoRR abs/1607.04606 (2016)." arXiv preprint arXiv:1607.04606 (2016).

2. The authors mention that they split the recordings into 3-second windows, with each window starting at word onset, and that baseline correction is applied to the neural data over the first 0.5 seconds of each window (lines 308–310). I would like to suggest a clarification regarding the baseline correction procedure. Typically, baseline correction involves subtracting the average signal from a period before the word onset, rather than from the initial part of the window following the word onset. Therefore, it would be more appropriate to apply baseline correction using the 0.5-second period preceding the word onset. This ensures that any neural activity directly related to the word processing is not inadvertently removed or altered by the baseline correction. Please confirm whether this adjustment was made or provide additional justification for the chosen method. Clarifying this point will help ensure the accuracy and interpretability of the results.

We appreciate this comment and agree that this procedure is non-standard. Motivated by this, we performed a comparison of decoding performance when the baseline correction is applied before (as opposed to during) the stimulus window in the supplementary material. As shown in the figure below, we did not notice major differences in decoding accuracy. We believe that this is because the deep learning model focuses on the relative spatio-temporal dynamics of M/EEG signals, as opposed to the absolute voltage or magnetic flux density.

We now report this result in the supplementary material, and state the following sentence in the main text:

“Note that while the baseline is usually computed on a segment of the data occurring before the stimulus, this did not change the decoding accuracies achieved, and hence we reuse data from the stimulus window for efficiency.”

3. The current study presents a deep learning pipeline based on MEG/EEG signals from different datasets, encompassing both reading and speech comprehension tasks. The results indicate that the model exhibits better decoding performance for MEG signals / reading tasks compared to EEG signals / speech comprehension tasks (Fig. 2D). However, it is not explicitly stated whether the authors trained separate models on each dataset individually or trained a single model on all datasets combined.

We apologize for this confusion, which was also highlighted by R3. We indeed trained the same architecture separately on each dataset, as we did not see any substantial gain in training in a single model across all datasets, as shown in the figure below.

We now report the comparison of separate-dataset versus joint-dataset training in Figure 8, and indicate in the main text:

“Importantly, we trained and tested the same architecture on each of these datasets separately. Training a single model jointly across multiple datasets did not lead to any performance boosts (see Fig 8). We speculate that the diversity of the datasets, which allows us to evaluate decoding performance across a variety of data regimes and experimental conditions, also hinders cross-dataset generalization. We leave the exploration of this challenge for future work.”

If the authors trained separate model on each dataset, it would be important to clarify how consistency was maintained across datasets in terms of data scale, signal quality, and other relevant factors. Ensuring such consistency is critical for avoiding discrepancies in decoding results attributable to variations in data quality or quantity.

R1 is right that experimental protocols vary across a variety of factors, including

- (1) the number of participants
- (2) the language and sentences used
- (3) the type of recording devices
- (4) the amount of data collected per participant,
- (5) the modality of stimulus presentation.

This heterogeneity is not only a challenge to the present study, but to the whole system neuroscience at large. At rare exceptions [1,2], experimental protocols greatly vary across studies.

To address this challenge, we collected and analyzed a new dataset, in which participants were presented with the exact same stimuli in the same recording machine (Le Petit Prince), either read or listened while their brain activity was recorded with the same MEG Megin System. However, even in this controlled setup, there are some discrepancies: for example, the reading task is slightly slower in the reading condition, as it was too difficult for subjects to read at the rate of normal speech.

However, we agree and amend the discussion to clarify this challenge:

“However, it is important to stress that the datasets presently analyzed differ across a wide variety of factors, such as number of subjects, precise tasks, type of words and languages. Consequently, future work remains necessary to ensure that the comparison across devices or experimental conditions hold true under minimally varying experimental protocols. This effort, recently initiated in the domain of image perception [1,2], would require participants to be presented to the very same sentences across a variety of tasks and experimental apparatus. The Little Prince MEG datasets presented here were acquired in both the reading and auditory modality to precisely contribute to a recent multi-site effort where participants from various countries are exposed to this book while their brain activity is recorded with various devices [3,4]. We hope that these datasets will contribute to clarify how differences across laboratories can be attributed to device, task and/or cognitive factors.

- [1] Hebart, Martin N., et al. "THINGS-data, a multimodal collection of large-scale datasets for investigating object representations in human brain and behavior." *Elife* 12 (2023): e82580.
- [2] Grootswagers, Tijn, et al. "Human EEG recordings for 1,854 concepts presented in rapid serial visual presentation streams." *Scientific Data* 9.1 (2022): 3.
- [3] Li, Jixing, et al. "Le Petit Prince multilingual naturalistic fMRI corpus." *Scientific data* 9.1 (2022): 530.
- [4] Momenian, Mohammad, et al. "Le petit prince hong kong (lpphk): Naturalistic fmri and eeg data from older cantonese speakers." *Scientific data* 11.1 (2024): 992.

4. The deep learning pipeline demonstrates commendable decoding performance across various datasets, with the overall decoding performance surpassing that of three baseline models: Linear, EEGNet, and BrainModule. However, I notice that for certain datasets, such as Nieuwland and LittlePrinceRead, the current model exhibits inferior performance relative to BrainModule (Fig. 2B). Could the authors explain why this discrepancy occurs? Specifically, exploring the reasons behind the reduced effectiveness of the proposed model on certain datasets would be beneficial. This could provide valuable insights into the strengths and limitations of the current approach and highlight areas for further improvement.

Thank you for highlighting this surprising effect. Indeed, the BrainModule alone outperformed our model (BrainModule+Transformer) in two datasets (especially the one from Newland et al, where subjects read de-contextualized sentences). The reasons for this remain unclear to us because of the heterogeneity of the protocols highlighted above.

5. The current manuscript shows that incorrect predictions share the same word length as the true word significantly more often than would be expected by chance (Fig. 5). Based on this observation, the authors argue that their decoding model captures visual features, specifically word length, during the training process (e.g., lines 466-469, and 492-502). I find this claim somewhat confusing. The current deep learning pipeline utilizes word embeddings generated by the t5-large model, which primarily focus on the semantic meaning and contextual relationships of words rather than their physical length. Moreover, word embeddings do not inherently encode any visual features of the words themselves. Given that word embeddings serve as the output target

in the current pipeline, it is unclear why the model would capture visual features in the training process. This point warrants further discussion and clarification from the authors.

Thanks for bringing up this interesting point. It is true that word embeddings are not designed to represent “visual” features of the words. However, their semantic features may statistically covary with such perceptual features. To clarify this point, we added the following analysis to the main text.

We ran a multivariate linear regression from the embeddings of the 5,000 most common English words to predict their number of letters. This yields a Pearson correlation of 0.71 ($p < 10^{-170}$) on the test set. Similarly, using a ridge classifier we can predict the first letter of the word embedding with an accuracy of 26%, where chance is 12% ($p < 10^{-11}$). This is described as follows in the main text:

“Note that these features are typically well encoded in the word embeddings: a linear regression trained to predict word length from the word embedding of the 5000 most common English words yields a Pearson correlation of 0.71 ($p < 10^{-170}$). Similarly, a ridge classifier trained on these word embeddings can predict the first letter of the word with an accuracy of 26% (chance: 12% $p < 10^{-11}$). This phenomenon is expected, as e.g. in English, ending in 'ed' generally marks past tense and starting with 'wh' generally marks a question.”

Minor:

1. The current study uses top-10 accuracy, which measures how often the correct word is within the model’s top 10 predictions. From a practical standpoint, this metric may be too lenient. Specifically, in real-world applications, users typically expect the correct prediction to be among the top few choices rather than being buried within a broader set of 10 options. Therefore, it is worth considering whether the top-10 accuracy provides an overly generous assessment of the model's performance. A more stringent evaluation, such as top-3 or top-5 accuracy, might better reflect the model’s practical utility. Additionally, discussing the implications of using top-10 accuracy for the model’s real-world applicability would be beneficial.

This is a valid concern. The top-10 accuracy is chosen because it is less noisy than top-1, which can unfairly penalize very close predictions, and therefore enables more reliable comparisons across experimental protocols.

Our revised paper includes the top-1 results in the first figure of the Supplementary Material. Note that we do emphasize in the limitations section the fact that, in spite of the significant improvements compared to state of the art, our method remains far from applicable in real-world settings.

We thank R1 for their thorough review and constructive feedback.

Reviewer #2

This study by Ascoli et al. proposes a deep learning pipeline to decode individual words from non-invasive EEG and MEG signals. They train and evaluate their pipeline with a large number of 723 participants and included different languages. Their analyses showed higher decoding performance compared to existing methods. This manuscript is straightforward in its design so the logic is easy to follow, and presented clearly both in the written and graphic elements. However, a number of points need clarifying and certain statements require further justification. My detailed comments are as follows:

The proposed methodology lacks a compelling demonstration of its novelty, as the authors did not explicitly delineate the distinctive features that differentiate their deep learning pipeline from existing approaches, particularly in relation to the model presented in the referenced prior study. A more rigorous comparative analysis is essential to substantiate the claim of a genuinely innovative contribution, rather than presenting what appears to be an incremental modification of established techniques. Regarding model selection, the rationale behind choosing the Transformer architecture remains unclear. The authors should provide a comprehensive theoretical justification that elucidates the architectural design choices.

We thank the reviewer for this comment.

Compared to Défossez et al (2023), there are many key differences.

Most importantly, the task is different: we aim to retrieve word embeddings rather than speech segments. This means that we do not require access to the ground truth speech segments at inference, a severe limitation towards real use-cases. We introduce a new contrastive loss function adapted to this new task, D-Siglip. Furthermore, on the architectural side, our model operates at the sentence level thanks to the transformer, which enables it to obtain significantly higher accuracies.

Finally, we validate our findings on nine different datasets, two of which are introduced in this paper and which use the same story (the Little Prince), the same MEG system (Megin 306 channels) but different tasks (Reading vs Listening).

We have emphasized these differences in the discussion section:

“Contrary to [1], our approach is based on word semantics, and does not require having access to the true spoken sounds in a retrieval set. Additionally, compared to the model used by [1] which operates on single words, our model operates at the sentence level thanks to the transformer, and is validated on a broader panel of datasets.”

[1] Défossez, Alexandre, et al. "Decoding speech perception from non-invasive brain recordings." *Nature Machine Intelligence* 5.10 (2023): 1097-1107.

I am also wondering whether the authors have tried directly encoding neural signals from MEG data as a neural signal encoder, then generating a text encoder directly for words, and

subsequently using the CLIP model. Could the authors elaborate on the specific advantages, computational strategies, and theoretical rationale behind mapping raw brain signals into a standardized word embedding space, particularly in terms of potential improvements in neural decoding accuracy, semantic representation fidelity, and cross-modal translation capabilities?

We did not try this specific approach.

The main advantages of our approach is that

- (1) It is strictly supervised, and thus provides a well-posed objective
- (2) It focuses on the word-level prediction, and can thus make use of a meaningful retrieval set that can be parametrically varied offline (i.e. the size of the decoded vocabulary)
- (3) The transformer still works at the sentence level to improve individual word predictions

We show that this approach outperforms Linear models, EEGNet and Défossez et al (2023)'s approach both quantitatively (Figure 1B), and qualitatively (our approach does not require knowing the speech sounds).

We evaluate the fidelity of semantic representations in Figures 4-6, and show that while decoding remains modest at the single trial levels, it gets surprisingly high semantic decoding once we average across several repetitions of the same word. We do not explore cross-modal translation.

Finally, using a standardized embedding space has the advantage of reusing representations (1) already established and interpreted within the NLP community (2) known to linearly align with the representations of the brain [1].

[1] Caucheteux, Charlotte, and Jean-Rémi King. "Brains and algorithms partially converge in natural language processing." *Communications biology* 5.1 (2022): 134.

I am also wondering why the repetition of the same word within a batch evoked different brain responses. Could it be due to semantic saturation or the dynamic allocation of attentional resources?

The same word can elicit different responses if it is presented to different subjects, or to the same subject but within different contexts.

Is a Multilingual Language Model trained and fine-tuned separately on each dataset? Are the training strategies consistent across different datasets? Have any adjustments been made to accommodate the specific characteristics of each dataset? Details should be added to address the above questions.

Thank you for bringing up this interesting point. T5 is a multilingual Language Model, and is used similarly across all datasets without finetuning.

This comment motivated us to re-run our experiments using another word embedding, namely from FastText [1]. As we describe in the Supplementary Material,

“FastText word embeddings are constructed by representing each word as a bag of character n-grams, which are then used to learn a weighted sum of these character sequences. The model is trained on a large corpus of text using a variant of the skip-gram model, with the goal of predicting a target word given its context words. FastText incorporates subword information by combining character n-grams and the word itself, allowing it to handle out-of-vocabulary words by generating embeddings from their character n-grams.”

In the figure below, we show that the average decoding performance across datasets displays a moderate drop from 23% to 20% top-10 accuracy when switching from T5 embedding to FastText embeddings. This result suggests that the choice of word embedding model is not critical when the goal is to decode non-contextualized word embeddings.

We report this figure in the Supplementary Material, and state the following in section 2.5:

“Interestingly, the particular model used to obtain word embeddings has a mild effect on decoding performance.”

[1] Bojanowski, Piotr, et al. "Enriching word vectors with subword information. CoRR abs/1607.04606 (2016)." arXiv preprint arXiv:1607.04606 (2016).

The authors did not explicitly elaborate on the semantic features extracted by the proposed model or provide a comprehensive mechanism for revealing these semantic characteristics. Further investigation is needed to elucidate the precise semantic representations and feature extraction processes underlying the model's decoding strategy.

Thank you for bringing up this point. Indeed, decoding is not an easy tool to interpret the neural code underlying neural signals: encoding tends to give more insight on these questions [1], although the increasingly large spaces of feature embeddings leave interpretability as a major challenge to both neuroscience and AI [2]. We thus now indicate in the discussion:

“However, the complexity of our decoding pipeline fundamentally limits our ability to understand how semantic representations are articulated and disentangled from syntactic and sensory features. The challenge of interpreting the neural activation patterns remains an open challenge for both neuroscience and AI [2], for which encoding models may be better suited than decoding models [1].”

[1] Naselaris, Thomas, et al. "Encoding and decoding in fMRI." *Neuroimage* 56.2 (2011): 400-410.

[2] Singh, Chandan, et al. "Rethinking interpretability in the era of large language models." *arXiv preprint arXiv:2402.01761* (2024).

The results show that the decoding performance is highly dependent on extensive individual-level data collection, which potentially indicates limitations in the model's ability to generalize neural representations across different participants, raising questions about the model's robustness and transferability of neural encoding patterns. These issues should be thoroughly addressed in this work.

These are good points which were also raised by other reviewers.

We have added the following discussion to address them:

“This result suggests that with a fixed recording budget, one should favor deep datasets (few participants over many sessions) than broad datasets (many participants over few sessions). This result is consistent with other decoding studies based on fMRI [1,2], and suggests that current experimental paradigms, typically based on the repetition of stimuli across participants, may limit the diversity of the training data and could hinder the decoding scalability across broad datasets. Finally, it is important to note that collecting a lot of data from a few individuals may limit other goals, such as the possibility to train a model that could decode language from the brain activity of unseen participants.”

Note also that while our model cannot generalize to unseen subjects in a zero-shot manner, it does exhibit cross-subject generalization as shown in Fig 3A where we see that test accuracy increases as we increase the number of participants in the training set. We now emphasize this further as follows:

“The results show that decoding performance increases with the amount of training data (Fig 3A), following a roughly log-linear trend. This result [...] demonstrates the effectiveness of the subject layers: they enable our model to generalize across participants, at the cost of preventing zero-shot generalization to new participants (for which the subject layers need to be fine-tuned).”

[1] Antonello, Richard, Aditya Vaidya, and Alexander Huth. "Scaling laws for language encoding models in fMRI." *Advances in Neural Information Processing Systems* 36 (2023): 21895-21907.

[2] Banville, Hubert, et al. "Scaling laws for decoding images from brain activity." *arXiv preprint arXiv:2501.15322* (2025).

We thank R2 for their thorough review and numerous feedback.

Reviewer #3

The authors developed a new decoder by adding a layer of transformer to CNN and attempted to decode the words from the brain signals using different publicly available databases as well as their own MEG recordings. The topic is now on a rise and attracts a lot of attention considering the potential use in BCI.

First of all, even though combining more than 700 subject recordings sounds very impressive, the abstract and their claim of developing the dataset is very misleading considering that they use mostly already available data and they didn't even combine the datasets. They train their model on each dataset separately and considering that the model is a black box it can adjust its parameters on every dataset.

The reviewer is right that we trained the model on each dataset separately.

We agree that the following sentence *"We train and evaluate our approach on an unprecedentedly large number of participants (723)"* can be misleading and thank the reviewer for pointing this out.

We have modified as follows:

"We evaluate our approach on seven public datasets and two datasets which we collect ourselves, amounting to a total of 723 participants"

Note however that we

- (1) used a single architecture
- (2) did not fit the model hyper parameters per dataset
- (3) train the model across multiple subjects
- (4) did not observe substantial gain in training across datasets.

In addition, our goal is not to interpret how the brain represents language, but to identify how words can be best decoded from M/EEG signals, including through data scaling. Black-box deep learning algorithms have repeatedly proved to be adequate for such a performance-driven objective. Overall, we believe that validating experimental findings on a large number of publicly available datasets is a useful endeavour to demonstrate the effectiveness of the proposed model.

We now report a comparison of separate versus joint training in the figure below.

We report this figure in the Supplementary Material and indicate in the main text:

“Importantly, we trained and tested the same architecture on each of these datasets separately. Despite our efforts to train a single model jointly across multiple datasets, we did not observe any performance boosts from such a procedure (see Fig 8). We speculate that the diversity of the datasets, which allows us to evaluate decoding performance across a variety of data regimes and experimental conditions, also hinders cross-dataset generalization. We leave the exploration of this challenge for future work.”

The other claim that it is better to have many recordings from a few subjects than vice versa is wrong. In that case the model will indeed perform better (unsurprisingly) on that dataset because intrasubject variability is always smaller than intersubject variability. But then one runs into the risk of overfitting the model. In that case the model and the obtained results will not possible to extrapolate on a larger population.

We appreciate this comment and would like to clarify these elements.

Our goal is not to build a model that would generalize to unseen subjects, but to maximize decoding performance in a given subject (i.e. the ability to decode unseen sentences for subjects belonging to the training set.). We agree with R3 that generalizing decoding to new subjects is likely to be a more difficult challenge.

We have added the following discussion:

“This result suggests that with a fixed recording budget, one should favor deep datasets (few participants over many sessions) than broad datasets (many participants over few sessions). This result is consistent with other decoding studies based on fMRI [1,2], and suggests that current experimental paradigms, typically based on the repetition of stimuli across participants, may limit the diversity of the training data and could hinder the decoding scalability across broad datasets. Finally, it is important to note that collecting a lot of data from a few individuals may limit other goals, such as the possibility to train a model that could decode language from the brain activity of unseen participants.”

Note also that while our model cannot generalize to unseen subjects in a zero-shot manner, it does exhibit cross-subject generalization as shown in Fig 3A where we see that test accuracy increases as we increase the number of participants in the training set. We now emphasize this further as follows:

“The results show that decoding performance increases with the amount of training data (Fig 3A), following a roughly log-linear trend. This result [...] demonstrates the effectiveness of the subject layers: they enable our model to generalize across participants, at the cost of preventing zero-shot generalization to new participants (for which the subject layers need to be fine-tuned).”

- [1] Antonello, Richard, Aditya Vaidya, and Alexander Huth. "Scaling laws for language encoding models in fMRI." *Advances in Neural Information Processing Systems* 36 (2023): 21895-21907.
- [2] Banville, Hubert, et al. "Scaling laws for decoding images from brain activity." *arXiv preprint arXiv:2501.15322* (2025).

The fact that read words were better decoded than heard ones was not surprising considering that the former ones are processed as one also based on their visual features, while the latter processed gradually as an auditory information arrives

To our knowledge, there is no systematic benchmark of word decoding from the auditory versus visual modality. However we agree that the RSVP may play a role in facilitating prediction since the words are better isolated from each other. Consequently, we now indicate:

“Mann Whitney tests across participants show that our decoding pipeline performs better when subjects read rather than listen to sentences. Indeed, the pairwise comparison of results for the Schoffelen and LittlePrince datasets, where the reading and listening stimuli were the same, yields $p < 10^{-16}$. Several reasons may explain this phenomenon. First, low-level visual features like word length are more readily represented and thus help decoding, as will be shown in what follows. Second, unlike the spoken words, which are not clearly segmented into low-level segments, the RSVP protocol may help isolate each word and thus improve our word-level decoding.”

Additionally you can find below some specific comments on the text:

Page 1, line 73: what is the performance accuracy?

Thank you for pointing this out, we added the result from the original paper (70% top-10 accuracy).

Page 5, line 308: did the baselining occur with the part of the trail that contains response to stimulus? As each window began with a word. It was 3s the duration of a sentence? That is not clear

Thank you for bringing this up. We agree that this procedure is non-standard. We added a comparison of decoding performance when the baseline correction is applied before versus during the stimulus window in the supplementary material, which is reported below.

We did not notice major differences in decoding accuracy when applying the baseline inside the stimulus window, rather than before as is commonly done. We believe that this phenomenon shows that our deep learning model focuses on the relative spatio-temporal dynamics of M/EEG signals, as opposed to the absolute voltage or magnetic flux density. We added the following sentence to clarify this:

“Note that while the baseline is usually computed on a segment of the data occurring before the stimulus, in our case this did not change the decoding accuracies achieved, and hence we reuse data from the stimulus window for efficiency.”

Page 6, line 343: is statistics corrected for multiple comparisons?

Thank you for raising this issue. We now add FDR correction for each model across the nine datasets considered, and this does not change our significance results, given the very low orders of magnitude of the p-values obtained (often below 10^{-10}).

Page 7, lines 371-372: type? Even though it doesn't feel like the accuracy difference would be significant

Thanks for catching this error. We now corrected this to:

“accuracy varies between 26 and 33% in the listening condition, and between 19 and 38% in the reading condition.”

Page 7, line 383-384: I don't understand this. Exposing subjects to the same sentence and comparing reading with listening is not the same

We apologize for the confusion. We only compare reading and listening results for the two datasets where the stimuli are the same. To clarify this issue, we now corrected the manuscript as follows:

“Indeed, the pairwise comparison of results for the Schoffelen and LittlePrince datasets, where the reading and listening stimuli were the same, yields $p < 10^{-16}$ ”.

Page 7, lines 392-394: then the results of the experiment cannot really be extrapolated to other individuals. Aren't we then running into the risk of overfitting the model? Did the author look at how the model did when the databases are combined? Given of course that the recording device (MEG or EEG) and the language are the same

This is a good point which was also made by other reviewers. We have added the following discussion to address it:

“This result suggests that with a fixed recording budget, one should favor deep datasets (few participants over many sessions) than broad datasets (many participants over few sessions). This result is consistent with other decoding studies based on fMRI [1,2], and suggests that current experimental paradigms, typically based on the repetition of stimuli across participants, may limit the diversity of the training data and could hinder the decoding scalability across broad datasets. Finally, it is important to note that collecting a lot of data from a few individuals may limit other goals, such as the possibility to train a model that could decode language from the brain activity of unseen participants.”

Note also that while our model cannot generalize to unseen subjects in a zero-shot manner, it does exhibit cross-subject generalization as shown in Fig 3A where we see that test accuracy increases as we increase the number of participants in the training set. We now emphasize this further as follows:

“The results show that decoding performance increases with the amount of training data (Fig 3A), following a roughly log-linear trend. This result [...] demonstrates the effectiveness of the subject layers: they enable our model to generalize across participants, at the cost of preventing zero-shot generalization to new participants (for which the subject layers need to be fine-tuned).”

[1] Antonello, Richard, Aditya Vaidya, and Alexander Huth. "Scaling laws for language encoding models in fMRI." *Advances in Neural Information Processing Systems* 36 (2023): 21895-21907.
[2] Banville, Hubert, et al. "Scaling laws for decoding images from brain activity." *arXiv preprint arXiv:2501.15322* (2025).

Page 8, line 432-437: what do the authors mean by “context”. In the methods section it seemed like they were averaging the words from the same sentence presented several times to the same subject. If it is so, the context is not different. Please clarify.

In the methods section, we distinguish between two different settings: averaging words belonging to the same sentence (for different subjects) versus averaging words belonging to different sentences (for a given subject), as illustrated in Figure 3C.

Page 9, line 503: what is the accuracy then? The figure is not very clear as the bar is too small. I don't understand the reasoning behind the statement that words outside the vocabulary are too rare. Why can't some frequent words be used in training and others in testing?

We added the score in the main text (6%), which is well above chance (1%). As we assign the train/test/validation split based on sentence chunks, enforcing that frequent words like "the" aren't in any of the sentences of the training set would remove an excessively large amount of training data.

Page 11, line 554-557: I don't think this claim can be done considering that there was no comparison with retrieval of non-words. Besides considering that the functional words has the highest accuracy this claim is even more doubtful

We believe that there is a misunderstanding here. What we mean is that the retrieval set is a fixed set of word embeddings. By contrast, Defossez et al's study makes use of retrieval across sound segments including, the true sound segment corresponding to the word considered (exactly as heard by the participant).

Besides the clarification of above mentioned specific comments, I could give the following suggestions to improve the article:

1. tone the conclusions and the abstract down considering that the dataset is not collected by the authors and the datasets from different papers are not merged

As discussed above, we have modified the abstract to address this valid concern.

2. I would try to train the model on one dataset and test on the other (given some words overlap in different datasets). If it performs better than other models, the added value of this model then will be shown.

Our approach does not allow such subject-generalization, because the subject layer is learnt end-to-end and thus cannot be applied to completely unseen subjects. Consequently, generalizing across datasets remains a challenging domain-adaptation problem which is beyond the scope of this study.

3. I missed in the methods the information on how much data was used for training and how much for testing. Was there an overlap between training and testing set?

We ensured that there would be no overlap between the train and test sets by splitting the texts at the sentence level, and assigning each sentence deterministically to a split (see section 2.6, paragraph “Splitting”).

Taking into account the above mentioned comments, my recommendation for this article in the current condition would be to not consider it for the publication.

We thank R3 for their thorough review and suggestions which helped improve the clarity of the present contribution.

Reviewer #4

The authors propose and evaluate a novel method to decode words from human electrophysiology data during language comprehension (total of 9 datasets, EEG or MEG). This was an impressive and computationally challenging undertaking, testing the robustness and scalability of language decoding from non-invasive brain recordings across 9 datasets of varying lengths and sizes, 3 indo-European languages, 2 task modalities (reading and listening), and 2 recording modalities (EEG and MEG).

This report contributes significantly to development and validation of methods in decoding non-invasively recorded brain signals. The improvements of their proposed pipeline, over reasonable baseline methods, are considerable (e.g. up to ~50% boost with their pipeline, translating to ~8% increase in accuracy, on average across datasets). Beyond accuracy boosts, this report thoroughly investigates the various factors and design decisions that influence brain decoding pipelines with the same experimental setup (number of words to decode from, dataset size, reading vs. writing). Typically studies will focus on one task or language at a time (e.g. just reading, English-only, just EEG etc.), so it's very valuable to see the evaluation of several datasets with common pipeline(s). It's an impressive effort in data curation and a good addition to the current literature.

I have no problems seeing this report published, but I do have a small number of mostly clarification points and editing suggestions that I would like to see addressed before I can recommend it for publication. I hope the authors find my comments sufficiently clear and useful in revising the manuscript. I believe the authors should not have too much trouble addressing my concerns.

Major

1. Preprocessing. The preprocessing section is quite short. I understand that given the extensive data curation, you kept the preprocessing steps to a smallest common denominator. I do wonder, however: were you concerned with data quality within datasets (artifact and bad channel removal) at all? Or were all datasets already preprocessed? Or did you simply fit to all channels and accept that bad channels would be dropped via poor decoding? If you had any heuristics in dealing with these issues, perhaps those would be useful to briefly document and briefly in an appendix or so (for readers that might attempt similar and are interested in pipeline development)?

This is a very good point. Indeed, we kept preprocessing to the minimum, as complex and manual steps are difficult to scale across datasets. In particular, we did not perform any artifact removal, which could indeed impact decoding performance. This decision was motivated by earlier work from Defossez et al (2023):

“[Defossez et al] study the effect of clamping and show that it is essential to ensure proper training. In Supplementary Section A.4, [they] further show that this approach is as effective as more complex data-cleaning procedures such as autoreject.”

We added a sentence to clarify this in the manuscript:

“Given the diversity of datasets considered, preprocessing was kept to a small common denominator: the M/EEG recordings were bandpass filtered between [0.1,40] Hz and resampled to 50 Hz, using built-in functions from MNE then scaled using sklearn’s RobustScaler and clamped in the range [-5,5]. We do not perform any artifact or bad channel removal in this work, relying on the deep learning model to discard any undesirable features.”

We now also added to the discussion:

“The present approach does not extensively search for ideal preprocessing parameters. We hope that the present benchmark will help identify these elements”

2. Deduplicated SigLIP. The section on your novel deduplication loss function was not entirely clear to me. You motivate the choice to use SigLIP loss in order to be able to deal with repeated words within a batch (which CLIP is not designed for). Then you state that for Deduplicated SigLIP you "discard the repeated words". Meaning you don't train the model on repetitions, always just one occurrence of the word in the batch -- does that mean that d-SigLIP is (in terms of loss signals used to train the classifier) effectively the same as CLIP (Based on the Fig. 8 you arrive at the schema for CLIP again?). Given that d-SigLIP gives you a decoding boost (table with Fig. 8), I understand it is a different loss, but I fail to gather from your description what distinguishes it from CLIP.

Also: How do you choose which elements of the batch are discarded? Do you just retain the first element of the batch and drop the rest or is it done some other way? Does your dropping criterion bias your classifier in any way (e.g. if you keep only the first element in the batch, would that mean that the model learns from brain signals at sequence onset)? In my opinion, it would be good to report that for completeness.

There is indeed a subtle difference between the two, and indeed the sentence “we discard the repetitions from the loss” is misleading: no word is fully discarded; rather some terms in the loss are masked. To clarify this issue we now indicate:

“Consider a batch size of N : for CLIP, this gives N “attractive terms” in the loss (green diagonal in fig 8) and N^2-N “repulsive terms” (red in fig 8). For SigLIP, if there are repeated words, there will be extra attractive terms towards their embeddings (the off-diagonal green elements in fig 8). What we do is mask out of the loss these extra attractive terms (blue in fig 8). This avoids excessively biasing towards frequent words.”

3. Baseline regression model time-point selection. On l. 170 you state: "We vary the offset of the time-point relative to the word onset between -0.5 and 2.5 seconds." On my first pass, I thought you were randomly selecting time-samples, but based on Fig 2A, I think it simply means you trained independent models per each sample point? I'd try to clarify that.

Indeed, independent models are trained for each sample point, following common practice in the literature [e.g. King et al, 2014]. We added the following sentence to clarify this:

"An independent model is trained for each timepoint between -0.5 and 2.5s, sampled at 50\,Hz."

[1] King, J. R., & Dehaene, S. (2014). Characterizing the dynamics of mental representations: the temporal generalization method. *Trends in cognitive sciences*, 18(4), 203-210.

4. Description of baseline models. I think it'd be helpful to add similar notation (dimensions of input features etc.) for baseline model description too, just like you do for the main pipeline. For example, are baseline regression models fit independently per M/EEG channel? Or were they fed all channels at the same time for each sample point? I would also move the more compact description of the baseline model from the results section (l 336-339) up the methods section (around line 165).

We have now reorganized as suggested by the reviewer, and clarified this point in the revised manuscript:

EEGNet and BrainModule are trained on all channels and all subjects of each study, in the same setting as our full pipeline.

5. Fig 2.G. On l. 387 you state: "This suggests that with a fixed recording budget, it is better to record a small number of participants across many sessions than a large number of participants with a small amount of sessions." This is an interesting analysis. Admittedly, it looks like the linear trend ($R = 0.7$) is likely driven by one high leverage point; perhaps that warrants qualification? We agree with the reviewer that this high leverage point limits the analysis. We now instead report the spearman correlation, which is not affected by the outlier, and yields $R=0.61$ ($p=0.038$), and specify this in the caption:

"The log-linear fit yields $p < 0.05$, and the Spearman correlation and p -value are reported above the figure."

In addition, note that this conclusion applies if one is solely concerned with accuracy. If one has a different goal in mind (e.g. generalization across subjects, say, developing a single decoder that works across subjects) that might change the consideration. Given that you are highlighting this result in the abstract, perhaps you can add a cursory comment about that too? Or do you think differently?

This is a good point which was also made by other reviewers.
We have added the following discussion to address it:

“This result suggests that with a fixed recording budget, one should favor deep datasets (few participants over many sessions) than broad datasets (many participants over few sessions). This result is consistent with other decoding studies based on fMRI [1,2], and suggests that current experimental paradigms, typically based on the repetition of stimuli across participants, may limit the diversity of the training data and could hinder the decoding scalability across broad datasets. Finally, it is important to note that collecting a lot of data from a few individuals may limit other goals, such as the possibility to train a model that could decode language from the brain activity of unseen participants.”

Note also that while our model cannot generalize to unseen subjects in a zero-shot manner, it does exhibit cross-subject generalization as shown in Fig 3A where we see that test accuracy increases as we increase the number of participants in the training set. We now emphasize this further as follows:

“The results show that decoding performance increases with the amount of training data (Fig 3A), following a roughly log-linear trend. This result [...] demonstrates the effectiveness of the subject layers: they enable our model to generalize across participants, at the cost of preventing zero-shot generalization to new participants (for which the subject layers need to be fine-tuned).”

[1] Antonello, Richard, Aditya Vaidya, and Alexander Huth. "Scaling laws for language encoding models in fMRI." *Advances in Neural Information Processing Systems* 36 (2023): 21895-21907.
[2] Banville, Hubert, et al. "Scaling laws for decoding images from brain activity." *arXiv preprint arXiv:2501.15322* (2025).

6. Fig 5A. Although, the analysis description is clear, I find parsing the figure somewhat difficult. What is the y-axis showing? How should it be read? Is it showing the % accuracy of for the target property? For example in Fig 5A, third bar (Schoffelen, green): does 40% accuracy mean that out of all test samples which are wrong (based on top-1 criterion), 40% of the time the decoded word will be the same length than the true word? Then, what do the hashed bars indicate? In addition, the label "Accuracy (failed predictions)" was not entirely intuitive to me. How can there be accuracy if a prediction is wrong? I was able to understand your textual recap of the results just fine, but looking at the figure I got confused and captions were not helpful. Perhaps I'm missing something obvious, but it would be helpful to spell out in captions what the bar hashing indicates, how to read the y-axis etc.

Your understanding is correct, and we have updated the label of the y axis to “proportion of failed predictions matching the target word property”. The hashed bars indicate chance level: we trained a dummy classifier on the same task but with labels shuffled.

7. Discussion. On l. 549. "This improvement is important to go beyond statistical metrics like the Pearson". It might just be a matter of wording, but why would a quantitative improvement in accuracy based on a new architecture speak to what metric one uses to test the model? If I read your work correctly, the message is that it is important to go beyond standard linear decoders (e.g. improving the loss objective, adding transformer modules for contextual effects etc.) rather than test metrics. Also, I don't see what is "statistical" about the Pearson correlation coefficients (compared to any other possible metrics). Might be helpful to clarify this statement a bit.

Thanks for bringing this up. What we mean is that linear decoders can significantly predict word embeddings above chance, as evaluated with an averaged metrics such as Pearson correlation. However, such an approach does not easily translate to real-word scenarios. However, we agree that this sentence is unclear and have removed it from the manuscript.

8. Discussion (l. 543). "Theoretically, it offers insights to the neural underpinning of language representations 545." I appreciate the attempt to add a theoretical angle to the discussion! But I do find this statement a bit of a stretch and the theoretical contributions here minimal (by necessity, given the strong methodological and engineering contributions, which I find totally fine). I agree that the work contributes to the long line of work using distributional models of word semantics and the cursory discussion of relevant past work is welcome. But distributional semantics is not tied to AI (i.e. artificial neural networks) as your discussion might suggest and you don't investigate competing lexical semantic approaches. Under "neural bases", I would expect a discussion/exploration of the role of different aspects of neural dynamics (spatio-temporal aspects, spectral power/phase etc.) in decoding which is beyond the scope of your work. I personally prefer concise statements of contributions (e.g. "Our work further shows the usefulness of distributional models of word semantics" ...). This just a suggestion.

We agree with this and have removed this vague statement.

We thank R4 for their thorough review and valuable feedback.

Rebuttal: Decoding words from noninvasive brain recordings

Reviewer #1 (Remarks to the Author):

All my concerns have been thoroughly addressed. The detailed responses provided, alongside the corresponding revisions to the manuscript, significantly enhance the clarity, rigor, and completeness of the work.

We thank the reviewer again for their help in improving our manuscript.

Reviewer #2 (Remarks to the Author):

The authors' revisions have significantly strengthened the manuscript through detailed clarifications, enhanced methodological transparency, and additional cross-dataset validation experiments, which collectively reinforce the rigor and generalizability of the findings. The inclusion of FastText embeddings, baseline protocol comparisons, and discussions on dataset heterogeneity further underscores the robustness of the proposed approach. However, a few issues still require further consideration to fully solidify the study's contributions:

1. While the authors emphasized differences from prior research (Défossez et al., 2023) (e.g., task objectives, contrastive loss functions, sentence-level Transformers), how did they quantify the practical impact of these differences? How to understand the quantitiveness represented by Figure 1B?

As indicated in the discussion, our approach enables direct word classification from the raw EEG/MEG data, whereas Defossez's approach consists in speech segment identification, which requires access to the ground truth speech segments being heard by the participants. Hence, from a practical point of view, this makes our method directly applicable to real-world decoding problems.

We quantitatively compared our model's results to those obtained with the exact architecture of Défossez et al in Figure 2B, trained using the same loss function as ours: this demonstrates a statistically significant impact of the sentence-level module. We additionally demonstrate the statistically significant impact of our loss function in Figure 11.

2. The authors attribute improved accuracy to "sentence-level operations" but omit critical details: Why is a Transformer more suitable for EEG/MEG time-series data than alternatives (e.g., CNNs, LSTMs)? How does self-attention specifically enhance alignment between neural signals and word embeddings? No analysis of attention patterns (e.g., visualizing which brain signal segments or frequency bands are prioritized) was provided.

We do use a CNN to handle the EEG/MEG time-series: the latter takes as input a single time-window of 3 seconds around word w (of dimension $n_{\text{channels}} \times n_{\text{timesteps}}$), and outputs a single vector $F(w)$ of dimension D , as illustrated in figure 1. Given a sentence with words w_0, \dots, w_N , the transformer is then applied on the sequence of vectors $[F(w_0), \dots, F(w_N)]$. The motivation here is that transformers are well-known to excel at contextual processing of words at the sentence-level. Note that the transformer does not exchange information between channels or time samples, but across words in the sentence, therefore the attention maps would not be informative of brain areas or frequency bands.

We leave the exhaustive comparison to other non-transformer architectures (LSTM, Mamba etc) to future research.

3. The authors attributed differences in brain responses to "different subjects or contexts" but provided no empirical evidence (e.g., neural signal visualizations, attention weight distributions, or correlation analyses). To strengthen this claim, the following should be added: a) Quantitative analysis of neural signal similarity across repeated words (e.g., Pearson correlation coefficients); b) Contextual effects on decoding performance (e.g., sentence position, semantic relatedness). Additionally, does the authors' explanation implicitly assume that the observed variability in neural responses is unrelated to potential mechanisms such as "semantic saturation" or "dynamic attentional allocation"?

Regarding a), we observe that averaging predictions, either across subjects or across word occurrences for a given subject, tends to improve decoding performance. This improved performance thus indicates that the patterns being averaged share a common representation. Regarding b), we already report a variety of quantitative analyses on (1) word frequency (2) word category (3) dataset. We also provide qualitative figures to explore semantic relatedness. We thus leave other interesting factors for future research to explore.

Regarding "semantic saturation" and "dynamic attentional allocation": These are interesting speculative proposals. However, understanding the full sources of variability in brain responses is beyond the scope of our current work.

4. The authors state that T5 embeddings were used uniformly across datasets without fine-tuning, but some questions remain:

How were cross-lingual discrepancies in neural signal distributions (e.g., syntax, phonology) handled? Were language-specific biases in pretrained T5 embeddings (e.g., English-centric training) mitigated?

These discrepancies were not handled: our model is trained separately on each dataset, therefore it learns the different distributions independently.

Language-specific biases were not corrected for either: to ensure maximal consistency across studies, we chose a single multilingual model to produce all word embeddings. Note that in Figure 10 of the revised manuscript, we replicated our findings with FastText embeddings, which are trained independently for each language, and observe very little differences compared to T5. This demonstrates that language-specific biases do not have a significant impact.

We thank the reviewer again for their help in improving our manuscript.

Reviewer #3 (Remarks to the Author):

I would like to thank authors for the extensive work they did on revising this article. My concerns are mostly addressed.

I would only like to emphasise that instead of saying that “(...) with fixed recording budget....” They should mention that choice for recording more data from less subjects should be guided by research question. For example, this approach might be more beneficial for bci studies where per subject performance improves with recording more data. However, as authors also mentioned, it will limit the interpretation of other goals as understanding the brain.

We thank the reviewer again for their help in improving our manuscript.

Reviewer #4 (Remarks to the Author):

I thank the authors for their response. They addressed all my concerns.

We thank the reviewer again for their help in improving our manuscript.